# DUET: OPTIMIZING LLM TRAINING DATA MIXTURES VIA NOISY FEEDBACK FROM UNSEEN EVALUATION TASKS

**Zhiliang Chen**[1,2], **Gregory Kang Ruey Lau** [1,3,*], **Chuan-Sheng Foo**[2] **& Bryan Kian Hsiang Low**[1]
[1]Department of Computer Science, National University of Singapore
[2]Agency for Research, Science, Technology and Research (A*STAR), Singapore
[3]CNRS@CREATE, 1 Create Way, #08-01 Create Tower, Singapore 138602
`{chenzhiliang, gregorylau}@u.nus.edu`
`lowkh@comp.nus.edu.sg`

## ABSTRACT

The performance of an LLM depends heavily on how well the training data matches the downstream evaluation task. However, in many practical settings, we typically do not know the data in the evaluation task (e.g., conversations between a chatbot and users are end-to-end encrypted). We refer to such tasks as *unseen evaluation tasks*. We can only deploy the LLM on these unseen evaluation tasks to gather multiple rounds of feedback on how well the model performs (e.g., gathering user ratings from a chatbot). In addition, this feedback can be noisy. How can we exploit such noisy feedback efficiently to optimize the LLM training data-mixture? Our paper presents **DUET**, a novel global-to-local algorithm that optimizes training data mixtures by interleaving *data selection* with *Bayesian optimization* to exploit coarse and noisy feedback from a downstream evaluation task. DUET is flexible enough to incorporate different data selection methods, each with different performance-compute tradeoffs. By analyzing DUET's *cumulative regret*, we theoretically show that DUET converges to the optimal training data mixture even without any fine-grained data information from an unseen task. Finally, our experiments across a variety of language tasks demonstrate that DUET attains substantial performance improvements over existing data selection and mixing methods in the unseen-task setting. Our library, which is flexible enough to optimize different LLM training ingredients, can be found at `https://github.com/chenzhiliang94/BO-for-LLMs`.

## 1 INTRODUCTION

The performance of an LLM depends heavily on how well the training data domain (Chen et al., 2024a; Xie et al., 2023a) matches the downstream evaluation task (Hoffmann et al., 2022; Long et al., 2017). For instance, if we knew that LLM users are interested in asking layman science questions, then training or fine-tuning the LLM with more Wikipedia data allows it to converse better with these users. Hence, knowing the evaluation task is important for curating a more relevant training data mixture, producing an LLM with better performance over the specific task of interest.

Unfortunately, in practice, the data (e.g., its domain, distribution, or labels) involved in an *unseen evaluation task* are often unknown. Thus, it is not obvious what data is relevant for training or fine-tuning the model. Consider the following problem setting: An LLM owner is interested in fine-tuning their LLM to converse better with users but due to privacy concerns (Li et al., 2024), conversations between the deployed LLM and users are end-to-end encrypted (`openai.com/enterprise-privacy`). Hence, the LLM owner does not know the actual evaluation data seen during test-time. Rather, they only receive coarse, noisy feedback on how well the LLM has performed in the conversation (e.g., user ratings or duration spent on the application). Of course, a naive idea is to simply iterate through all possible data mixtures and observe the resulting LLM performance, which is too computationally expensive. A better question is to perhaps ask: How can we exploit the noisy feedback loop efficiently to improve and optimize the training data mixture?

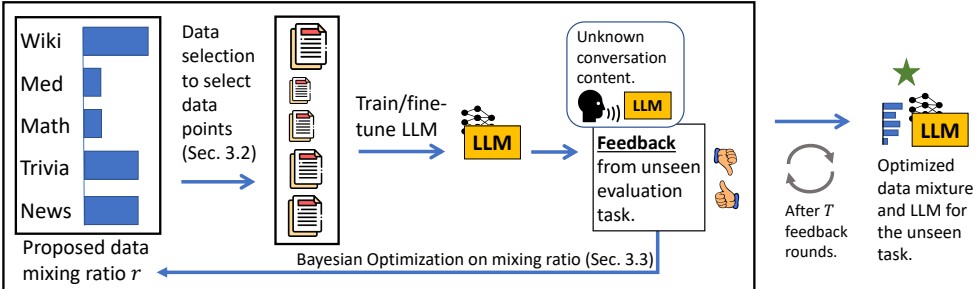

Figure 1: **DUET** exploits a feedback loop to optimize the data mixture for an unseen evaluation task. In contrast, conventional data mixing and selection works require fine-grained data information of the task, which is not available here.

This paper presents **DUET** (Fig. 1), an efficient algorithm that exploits a noisy feedback loop to optimize the training **D**ata mixture for an **U**nseen **E**valuation **T**ask. DUET is a *global-to-local* algorithm that interleaves *data selection* (Albalak et al., 2024; Ting & Brochu, 2017; Koh & Liang, 2017) with *Bayesian optimization* (BO) (Snoek et al., 2012; Srinivas et al., 2010) to optimize the training data mixture. Globally, BO in DUET uses coarse, noisy feedback from the unseen evaluation task to automatically refine the mixing ratio of data domains in the training data mixture iteratively. Locally, DUET uses data selection to retrieve high-quality data points from each data domain until the proposed mixing ratio is reached. This results in an algorithm that can optimize training data iteratively even without having access to fine-grained data information from the evaluation task.

**Related works.** In our problem setting, (a) there is no direct access to the data (e.g., its domain, distribution, or labels) involved in the unseen evaluation task but (b) we can gather multiple rounds of feedback (details covered in Sec. 2.2) from the task using an LLM. App. A.1 provides a few more practical examples of this setting. This setting is different from those considered in conventional domain adaptation (DA) and domain generalization (DG) works. Prior DA works assume fine-grained knowledge of data (e.g., labeled/unlabeled data (Zhang et al., 2022) or data distribution (Ganin & Lempitsky, 2015; Zhang et al., 2021)) from the evaluation task for selecting relevant training data that match the evaluation data. On the other hand, DG considers a rigid setting with no knowledge (not even feedback) of the evaluation task (Muandet et al., 2013; Shin et al., 2024; Wang et al., 2022).

Similarly, *data mixing* works such as DoReMi (Xie et al., 2023a), BiMix (Ge et al., 2025) and more (Chen et al., 2024a; Fan et al., 2024; Xie et al., 2025; 2023b) introduced methods to optimize data mixtures, and *data selection* works (Albalak et al., 2024; Xia et al., 2024; Pruthi et al., 2020; Xie et al., 2023b) explored ways to find high-quality data to improve an LLM's performance. However, these methods assume some availability of fine-grained evaluation data information, such as evaluation gradients, labels, distribution or naively assuming the training data shares the same distribution as the task. In practice (like in our setting), these are not always available. In fact, when we applied existing data mixing and selection methods directly to our setting, they perform worse than DUET (Sec. 6). We provide more discussion of the shortfalls of these prior works in App. A.2.

To the best of our knowledge, DUET is the first work that interleaves data selection with BO to iteratively optimize training data mixture based on feedback from an unseen evaluation task. At first glance, eliciting multiple rounds of feedback with BO seems expensive. However, BO is sample-efficient and is the only way we can exploit such coarse and noisy feedback iteratively, unlike prior methods that require much more fine-grained data information (see above). In fact, subjecting models to multiple rounds of training or fine-tuning in a feedback loop is a natural part of the deployment life-cycle to improve LLMs. Specifically, our contributions are:

- We introduce a novel and realistic problem setting where the data involved in an unseen evaluation task is unknown but we can deploy our LLM to gather multiple rounds of coarse and noisy feedback. Then, we introduce **DUET**, a novel algorithm that exploits the feedback loop to optimize training **D**ata mixture for the **U**nseen **E**valuation **T**ask. To achieve this, DUET interleaves data selection (Sec. 3.2) with Bayesian optimization (Sec. 3.3) to iteratively optimize the training data mixture. DUET is flexible enough to incorporate any data selection choice in its inner loop, and we qualitatively and quantitatively analyzed different choices in our paper.

- We provide a theoretical analysis of DUET's convergence to the optimal training data mixture by analyzing DUET's *attained cumulative regret* (Chen et al., 2024b; Chowdhury & Gopalan, 2017) under the BO framework (Sec. 4).
- We demonstrate the effectiveness of DUET on LLM fine-tuning for language tasks comprising both in-domain and out-of-domain unseen tasks spanning different domains. Compared to conventional data selection and mixing methods (e.g., DoReMi, LESS, Aioli (Chen et al., 2024a) and more), DUET produces more optimal training data mixtures (Sec. 6.2).

## 2 PRELIMINARIES

### 2.1 BAYESIAN OPTIMIZATION

We first provide an outline of how BO can be used to optimize a generic black-box objective function before explaining how BO is used in DUET (Sec. 3.3). We consider a black-box objective function $f : \mathbb{R}^n \mapsto \mathbb{R}$ over the space of inputs $r \in \mathbb{R}^n$. As we show later (Sec. 2.2), we will use the data mixing ratio as $r$ in our setting. The goal is to find $r^* \triangleq \arg\min_r f(r)$ which minimizes the objective function. BO is a query-efficient *active algorithm* that strategically selects input points to query the black-box objective function, conditioned on previous function observations. At each iteration $t = 1, 2, \ldots, T$ of BO, we query the black-box function with a selected input $r_t$ to obtain a *noisy* observation $\tilde{y}_t \triangleq f(r_t) + \epsilon_t$ with a sub-Gaussian noise $\epsilon_t$ (e.g., Gaussian or bounded noise) to form sample $(r_t, \tilde{y}_t)$. Consistent with Chowdhury & Gopalan (2017), we model the unknown function $f$ as a realization of a *Gaussian process* (GP) (Williams & Rasmussen, 2006) that is fully specified by its *prior* mean $\mu(r)$ and covariance $\kappa(r, r')$ for all $r, r' \in \mathbb{R}^n$ where $\kappa$ is a *kernel* function chosen to characterize the correlation of the observations between any two inputs $r$ and $r'$; a common choice is the *squared exponential* (SE) kernel $\kappa(r, r') \triangleq \exp(-\|r - r'\|_2^2/(2m^2))$ with a *length-scale* hyperparameter $m$ that can be learned via maximum likelihood estimation. Given a column vector $\boldsymbol{y}_t \triangleq [\tilde{y}_\tau]_{\tau=1,\ldots,t}^\top$ of noisy observations at previous inputs $r_1, \ldots, r_t$, the posterior belief of $f$ at any new input $r'$ is a Gaussian distribution with the following *posterior* mean and variance:

$$
\begin{aligned}
\mu_t(r') &\triangleq \kappa_t^\top(r')(K_t + \zeta I)^{-1}\boldsymbol{y}_t \\
\sigma_t(r') &\triangleq \kappa(r', r') - \kappa_t^\top(r')(K_t + \zeta I)^{-1}\kappa_t(r')
\end{aligned}
\tag{1}
$$

where $\kappa_t(r') \triangleq [\kappa(r', r_\tau)]_{\tau=1,\ldots,t}^\top$ is a column vector, $K_t \triangleq [\kappa(r_\tau, r_{\tau'})]_{\tau, \tau' \in 1, \ldots, t}$ is a $t \times t$ covariance matrix, and $\zeta > 0$ is viewed as a free hyperparameter that depends on the problem setting (Chowdhury & Gopalan, 2017). Using equation 1, the BO algorithm selects the next input query $r_{t+1}$ by optimizing an *acquisition function*, such as minimizing the *lower confidence bound* (LCB) acquisition function (Srinivas et al., 2010): $r_{t+1} = \arg\min_r \mu_t(r) - \beta_{t+1}\sigma_t(r)$ with an exploration parameter $\beta_{t+1}$. In addition, BO can also handle constraints on inputs $r$ (Gardner et al., 2014). The cumulative regret (for $T$ BO iterations w.r.t. a minimization problem) $R_T \triangleq \sum_{t=1}^T [f(r_t) - f(r^*)]$ is used to assess the performance of a BO algorithm (Tay et al., 2023) given that $f(r^*)$ is the true function minimum. A lower cumulative regret indicates a faster convergence rate. We provide a theoretical analysis of DUET's cumulative regret in Sec. 4.

### 2.2 PROBLEM SETTING: OPTIMIZING DATA MIXTURES FOR AN UNSEEN TASK

Now, we formally describe our problem setting. Suppose that we have $n$ training datasets $\mathcal{D} \triangleq \{D_1, D_2, \ldots, D_n\}$ from $n$ different domains (e.g., Wikipedia, ArXiv), where $\mathcal{D}$ is the union of these training datasets. Let $\mathcal{L}_{\text{eval}}(\theta)$ be the unseen evaluation task loss w.r.t. an LLM parameterized by $\theta$. This "loss" represents feedback from the unseen evaluation task and does not have a closed, mathematical form. Our goal is to find an optimal data mixture $\mathcal{X}^* \in \mathcal{D}$ (a set of training data points) and learn model parameters $\theta_{\mathcal{X}^*}$ such that the unseen evaluation task loss $\mathcal{L}_{\text{eval}}$ is minimized:

$$
\begin{aligned}
\min_{\mathcal{X} \in \mathcal{D}} \quad & \mathcal{L}_{\text{eval}}(\theta_{\mathcal{X}}) \\
\text{s.t.} \quad & |\mathcal{X}| = M,
\end{aligned}
\tag{2}
$$

where $\theta_{\mathcal{X}} \triangleq \arg\min_\theta \mathcal{L}_{\text{train}}(\mathcal{X}, \theta)$ is the model parameters learned in a standard supervised learning manner (e.g., gradient descent) from a chosen data mixture $\mathcal{X}$ and $\mathcal{L}_{\text{train}}$ is a standard model training

loss (e.g., cross-entropy loss for LLM prediction). To make our theoretical formulation and expository simpler, we consider the feedback $\mathcal{L}_{eval}$ deterministic. However, DUET works equally well for in noisy feedback setting, which we demonstrate empirically (Sec. 6) and elaborate in App. A.3. $M$ is a practical, pre-decided constraint (Mirzasoleiman et al., 2020) to ensure the selected data mixture is not too large. In practice, evaluation task loss $\mathcal{L}_{eval}$ is just a feedback that indicates how well the LLM is performing and does not contain any evaluation data information. It can also be interchanged with other measures to be maximized (e.g., accuracy, user ratings) with slight mathematical adjustment to later statements.

# 3    OPTIMIZING TRAINING DATA MIXTURES USING DUET

Unfortunately, solving problem 2 is challenging because the unseen evaluation task loss $\mathcal{L}_{eval}$ does not have a closed, mathematical form and finding the optimal data mixture $\mathcal{X}^*$ directly is a high-dimensional discrete optimization problem. To address this, DUET adopts a global-to-local approach to optimize the training data mixture. Globally, DUET uses BO to adjust the mixing ratio in the data mixture adaptively based on the task feedback. Locally, we interleave a data selection method of choice (depending on the practitioner's compute budget) to refine the data mixture every iteration. Fig. 2 illustrates, in a simple setting, how DUET progressively finds better data mixtures close to the optimal (green star). We also discuss several extensions of DUET in App. A.3.

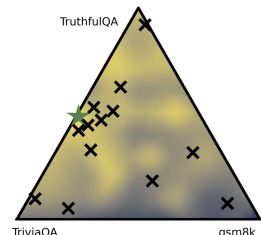

Figure 2: DUET finds the optimal data mixture iteratively and strategically.

## 3.1    REPARAMETERIZATION OF THE OPTIMIZATION PROBLEM

We first reparameterize the objective function of problem 2 into a bilevel optimization problem that, at the outer level, depends on the mixing ratio $r \in \mathbb{R}^n$ of training data domains (such reparameterization has been considered in AutoML works (Chen et al., 2024b)). This reparameterized problem has a unique structure that aligns with DUET's global-to-local nature (Sec. 3.2 & 3.3).

**Theorem 3.1.** $\mathcal{X}^*$, the optimal set of data points from $\mathcal{D}$, is the solution of the original problem 2 iff $r^* = ratio(\mathcal{X}^*)$ is the optimal mixing ratio solution of the reparameterized problem:

$$\min_{r \in \mathbb{R}^n} \min_{\mathcal{X} \in S_r} \quad \mathcal{L}_{eval}(\theta_{\mathcal{X}}), \tag{3}$$

where $S_r \triangleq \{\mathcal{X} : \mathcal{X} \in \mathcal{D}, ratio(\mathcal{X}) = r, |\mathcal{X}| = M\}$ and $ratio(\mathcal{X}) = r$ means that the data points in $\mathcal{X}$ satisfies the mixing ratio $r \in \mathbb{R}^N$ from $n$ data domains and $\|r\|_1 = 1$.

The proof can be found in App. B.1, where we show that $\mathcal{X}^*$, the optimal data mixture of original problem 2, satisfies a mixing ratio $r^*$ that is also the solution of reparameterized problem 3. DUET aims to solve problem 3 in an iterative manner. At the outer optimization level (global), DUET uses BO to exploit feedback from the evaluation task to propose a promising mixing ratio $r_t$ at each iteration $t$. At the inner optimization level (local), we introduce a sampling strategy that uses local domain data selection to retrieve a high-quality data subset that satisfies mixing ratio $r_t$.

## 3.2    USING DATA SELECTION METHODS FOR INNER PROBLEM

In this section, we show how data selection methods can be used to solve the inner problem in DUET. **For ease of expository and illustration, we use Influence Function (IF) as the choice of data selection method to explain our method.** DUET is flexible enough to incorporate different data selection choice and we analyzed different data selection methods in our experiments (Sec. 6). Our inner optimization problem aims to find the best-performing data mixture that satisfies:

$$\mathcal{X}_r^* \triangleq \arg\min_{\mathcal{X} \in S_r} \quad \mathcal{L}_{eval}(\theta_{\mathcal{X}}), \tag{4}$$

where $S_r \triangleq \{\mathcal{X} : ratio(\mathcal{X}) = r, |\mathcal{X}| = M\}$. In other words, we need to find a subset of data $\mathcal{X}_r^*$ that yields the lowest evaluation task loss $y_r^* = \mathcal{L}_{eval}(\theta_{\mathcal{X}_r^*})$ while constrained to mixing ratio $r$.

First, let's consider a simple approach, based on prior works on estimating distribution extrema (de Haan, 1981; Lee & Miller, 2022). We *randomly* sample $k$ different data mixtures from $S_r$. This yields $k$ data mixture samples $\{\mathcal{X}_1, \ldots, \mathcal{X}_k\}$ (each satisfying the mixing ratio $r$). A **uniform random estimator** for $y_r^*$ is obtained by evaluating the unseen task performance of an LLM trained on each data mixture sample and taking the minimum: $\widetilde{y}_r^* = \min_{\mathcal{X}_i}\{\mathcal{L}_{\text{eval}}(\theta_{\mathcal{X}_1}), \ldots, \mathcal{L}_{\text{eval}}(\theta_{\mathcal{X}_k})\}$ with $\widetilde{\mathcal{X}}_r^* = \arg\min_{\mathcal{X}_i}\{\mathcal{L}_{\text{eval}}(\theta_{\mathcal{X}_1}), \ldots, \mathcal{L}_{\text{eval}}(\theta_{\mathcal{X}_k})\}$ as the solution estimate of inner problem 4. The estimator $\tilde{y}_r^*$ is the 1st-order statistic (Arnold et al., 2008) and a random variable. While consistent (i.e., as we increase the sampling size $k$, we can estimate the solution of Eq. 4 more accurately), uniform random estimator $\tilde{y}_r^*$ has high variance (we provide empirical evidence in Fig. 8) because from $k$ uniformly random data mixture samples, it is unlikely we select the optimal data mixture.

**How can data selection help?** We aim to improve the quality of estimator $\widetilde{y}_r^*$ by incorporating data selection methods (Sim et al., 2022; Wang et al., 2024a) into our sampling process. Specifically, we want to increase the chance of sampling high-quality data points (conversely, reduce the chance of sampling low-quality data points) from each data domain, before using it to train an LLM. To do so, let us consider the use of influence function (Koh & Liang, 2017; Saunshi et al., 2023) (IF) as a data selection method into our estimator $\widetilde{y}_r^*$ to estimate the inner problem solution more accurately. In App. A.5, we discuss the tradeoffs between different data selection methods, such as coresets (Mirzasoleiman et al., 2020), diversity-driven measures (Wang et al., 2024b) and LESS (Xia et al., 2024) when used in DUET. Our experimental results (Fig. 6) also analyzed the performance of DUET paired with different data selection methods.

**IF-driven estimator**. We construct an IF-driven estimator in the following manner: *first*, for each dataset $D_i \in \mathcal{D}$ from the training domains, we fine-tune a separate, potentially smaller, LLM on that dataset. *Second*, we derive the IF score of each training data point w.r.t. the trained LLM for its respective domain (this can be computed and stored beforehand; more details in App. A.4). *Lastly*, given a mixing ratio $r$ proposed at each iteration, we perform weighted sampling from each domain based on each data point's IF score within the domain dataset (instead of uniform sampling as mentioned previously) until we satisfy the mixing ratio $r$. From hereon, we refer to this sampling process as *IF-weighted sampling*. For each data domain, there is a higher chance to sample a data point with a higher IF score. This yields a data mixture sample $\mathcal{X}^{IF}$. By performing IF-weighted sampling $k$ times, we obtain $k$ samples of IF-weighted data mixtures $\{\mathcal{X}_1^{IF}, \ldots, \mathcal{X}_k^{IF}\}$, producing a new **IF-driven estimator**:

$$\widetilde{y}_r^* = \min_{\mathcal{X}_i}\{\mathcal{L}_{\text{eval}}(\theta_{\mathcal{X}_1^{IF}}), \ldots, \mathcal{L}_{\text{eval}}(\theta_{\mathcal{X}_k^{IF}})\}, \tag{5}$$

which estimates the solution of inner optimization problem 4. Unlike the uniform random estimator mentioned earlier, IF-driven estimator emphasizes selecting data with high IF scores, and prior works (Saunshi et al., 2023) have regarded data points with higher IF scores as of higher quality. Next, we will discuss the empirical distribution of the IF-driven estimator.

**Empirical distribution**. In Fig. 3, we mixed data from two training domains to train an LLM to maximize an unseen task accuracy (while Eq. 4 & 5 consider the minimization case, we can use $\max$ instead of $\min$ for the maximization case). We used a fixed mixing ratio $r = [0.5, 0.5]$. The optimal data mixture satisfying this ratio attains a task accuracy indicated by the red line (obtained by iterating through all possible data mixtures in a brute-force manner). Ideally, we want our estimator to be as close to the red line as possible. Next, we plot the empirical distribution of the **uniform random estimator** and **IF-driven estimator**. Empirically, the IF-driven estimator (green histogram) has a lower variance and bias than the uniform random estimator (gray histogram), producing a closer estimate to the true solution (red line). This suggests that the IF-driven estimator $\widetilde{y}_r^*$ estimates the solution of problem 4 more accurately.

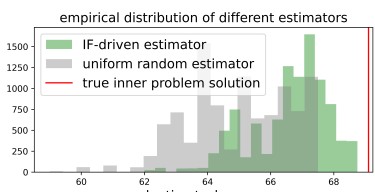

Figure 3: Empirical distribution of the uniform random and IF-driven estimator $\widetilde{y}_r^*$. Red line is the true inner problem solution.

**Theoretical distribution**. Exactly how well does the IF-driven estimator $\widetilde{y}_r^*$ estimate the optimal unseen evaluation task loss $y_r^*$ w.r.t. a given data ratio $r$? To answer this, we theoretically analyze this estimator's empirical distribution. Empirically (App. A.7), the negative of the sampling distribution of the unseen task accuracy (we consider the negative because we are looking to maximize the accuracy,

instead of minimizing the loss) of each sample $\mathcal{L}_{\text{eval}}(\theta_{\mathcal{X}^{IF}})$ resembles a truncated exponential distribution. Based on this, we characterize how well the IF-driven estimator $\widetilde{y_r^*}$ estimates $y_r^*$:

**Theorem 3.2.** *Let* $\{\mathcal{X}_1^{IF}, \ldots, \mathcal{X}_k^{IF}\}$ *be* $k$ *data mixture samples drawn from* $S_r$ *using IF-weighted sampling. Furthermore, assume each independent sample* $\mathcal{L}_{\text{eval}}(\theta_{\mathcal{X}_i^{IF}})$ *follows the shifted truncated exponential distribution* $y_r^* + \exp_t(\lambda, c)$, *for* $i = 1, 2, \ldots, k$ *where* $\exp_t(\lambda, c)$ *is a truncated exponential distribution governed by rate parameter* $\lambda$ *and truncated at* $c > 0$. *Then, the IF-driven estimator* $\widetilde{y_r^*}$ *defined in Eq. 5 is a random variable:* $y_r^* + \epsilon$, *where* $y_r^*$ *is the true inner problem solution of Eq. 4 and* $\epsilon$ *is a random noise variable with probability density function (PDF):*

$$PDF_\epsilon(u) = \frac{\lambda k e^{-\lambda u}}{1 - e^{-\lambda c}} \left( \frac{e^{-\lambda u} - e^{-\lambda c}}{1 - e^{-\lambda c}} \right)^{k-1} \text{ on } u \in [0, c] .$$

The proof is shown in App. B.2 and computes the probability distribution of the 1st order statistic (in which our estimator uses) of a truncated exponential distribution. Theorem 3.2 is used in DUET's convergence analysis in Sec. 4. In App. B.4, we also provide details to help readers extend our analysis to other empirical sampling distributions. This also indicates that estimation error $\epsilon$ of the IF-driven estimator reduces to 0 as the sampling size $k$ increases asymptotically. Surprisingly, our experiments (Sec. 6) show that using $k = 1$ is enough to select good data mixtures, underscoring the effectiveness of using data selection as opposed to random sampling. We also found that given sufficient budget, using varying $k$ gives us granular control of DUET's performance (Sec. 6.3).

### 3.3 USING BAYESIAN OPTIMIZATION FOR OUTER PROBLEM

With the IF-driven estimator introduced to estimate the inner optimization problem solution (or any estimator using a desired data selection method of choice), we shift our focus to solving the outer optimization problem of problem 3, which aims to find the optimal data mixing ratio $r^*$ for the unseen evaluation task. Since the solution of the inner problem $y_r^* = \min_{\mathcal{X} \in S_r} \mathcal{L}_{\text{eval}}(\theta_{\mathcal{X}})$ depends only on the mixing ratio $r$, we can define a function $f(r) \triangleq y_r^* = \min_{\mathcal{X} \in S_r} \mathcal{L}_{\text{eval}}(\theta_{\mathcal{X}})$, where for a given mixing ratio $r$, we use the IF-driven estimator to estimate a solution for the inner problem, producing $f(r)$. As such, the outer optimization problem of problem 3 can be rewritten into $\min_r f(r)$ where $r \in \mathbb{R}^n$ is a probability simplex representing the mixing ratio over the $n$ training domains. DUET uses BO constrained to $\|r\|_1 = 1$ (Sec. 2.1) to find the optimal mixing ratio $r^*$ for the outer problem.

BO is suitable for solving this problem for a few reasons. First, evaluating $f$ requires us to use the IF-driven estimator to estimate the inner optimization problem solution and thus $f$ is a black-box function with no closed, mathematical form; BO is a principled and popular framework to optimize such black-box functions (Garnett, 2023; Pyzer-Knapp, 2018). Second, we are estimating the inner problem solution (Theorem. 3.2) using data selection. This implies we can only obtain *noisy observations* $f(r) + \epsilon$, where $\epsilon$ is a random noise variable with the same distribution as that in Theorem 3.2; fortunately, BO handles noisy function observations gracefully (Srinivas et al., 2010; Chowdhury & Gopalan, 2017) during the optimization process, allowing us to find the optimal mixing ratio eventually (theoretical results shown in Sec. 4).

### 3.4 INTERLEAVING THE IF-DRIVEN ESTIMATOR AND BO

DUET uses BO at the outer level and IF-driven estimator at the inner level to iteratively optimize the data mixture, solving problem 3. We formally describe DUET in Algorithm 1.

At iteration $t$, DUET uses the LCB acquisition function (Srinivas et al., 2010) on the GP posterior to propose a candidate mixing ratio $r_t$ for our data domains (Line 3). Using the proposed mixing ratio $r_t$, we use IF scores of each data point to compute the IF-driven estimator $\widetilde{y_{r_t}^*}$ and fine-tune an LLM with the selected data points and observe the feedback from the downstream unseen task based on the fine-tuned LLM. We keep track of the best performing data mixture sample $\mathcal{X}_t^*$ at every iteration $t$ (Line 4, 5 and 6). Next, we include $(r_{t+1}, \widetilde{y_{r_t}^*})$ into our historical observations $\mathcal{D}_{t+1}$ (Line 7) and update our GP posterior (Line 8). After which, we repeat the entire feedback process to select the next LLM fine-tuning data mixture, until the budget of $T$ BO iterations is exhausted. In the end, we recover the best performing data mixture $\mathcal{X}^*$ for the unseen evaluation task (Line 10).

---

**Algorithm 1 DUET**: Optimizing training **D**ata Mixtures for an **U**nseen **E**valuation **T**ask

---

1: **Input:** $n$ training datasets from $n$ domains $\{D_1, \ldots, D_n\}$. Computed IF scores of each data point (App. A.4) w.r.t. its domain dataset and locally trained model. Initial observation of data mixing ratio and evaluation task performance: $\mathcal{D}_0 \triangleq \{(r_0, \tilde{y}_0)\}$, SE kernel $\kappa$, sampling size $k$, parameter $\beta_t$ for acquisition step and total number of BO iterations $T$.
2: **for** $t = 1, \ldots, T$ **do**
3:     $r_t = \arg\min_r \mu_t(r) - \beta_t \sigma_t(r)$ (BO acquisition step)
4:     IF-weighted sampling to obtain $k$ samples of data mixtures $\{\mathcal{X}_1^{IF}, \ldots, \mathcal{X}_k^{IF}\}$ (Sec. 3.2).
5:     **IF-driven estimator** at iteration $t$:
      $\widetilde{y_{r_t}^*} = \min_{\mathcal{X}_i} \{\mathcal{L}_{\text{eval}}(\theta_{\mathcal{X}_1^{IF}}), \ldots, \mathcal{L}_{\text{eval}}(\theta_{\mathcal{X}_k^{IF}})\}$.
6:     Keep track of best performing data mixture $\mathcal{X}_t^* = \arg\min_{\mathcal{X}_i} \{\mathcal{L}_{\text{eval}}(\theta_{\mathcal{X}_1^{IF}}), \ldots, \mathcal{L}_{\text{eval}}(\theta_{\mathcal{X}_k^{IF}})\}$.
7:     $\mathcal{D}_t = \mathcal{D}_{t-1} \cup \left\{\left(r_t, \widetilde{y_{r_t}^*}\right)\right\}$
8:     Update the GP posterior and $\kappa$ with updated observations $\mathcal{D}_{t+1}$ (Sec. 2.1).
9: **end for**
10: $\mathcal{X}^* = \arg\min_{\mathcal{X}_i^* \in \{\mathcal{X}_1^*, \ldots, \mathcal{X}_T^*\}} \mathcal{L}_{\text{eval}}(\theta_{\mathcal{X}_i^*})$

---

## 4 THEORETICAL ANALYSIS

### 4.1 CONVERGENCE ANALYSIS OF DUET USING CUMULATIVE REGRET

We analyze the convergence rate of DUET using the growth of *attained cumulative regret* (Chen et al., 2024b) $\tilde{R}_T = \sum_{t=1}^T |\widetilde{y_{r_t}^*} - f(r_t)| = \sum_{t=1}^T |f(r^*) + \epsilon_t - f(r_t)|$ for $T$ BO iterations. The attained cumulative regret consists of two terms, where $|f(r^*) - f(r_t)|$ indicates the quality of mixing ratio $r_t$ proposed at each iteration while $\epsilon_t$ indicates how well we can estimate the inner problem solution at every iteration. By analyzing the attained *average* regret $\tilde{R}_T/T$ with $T \to \infty$, the following Theorem helps us understand how close our algorithm converges (Berkenkamp et al., 2019).

**Theorem 4.1.** *Let $f$ be the outer problem objective defined in Sec. 3.3 with bounded RKHS norm: $||f||_\kappa = \sqrt{\langle f, f \rangle_\kappa}$. Also, let our IF-driven estimator for the inner problem solution be governed by the error distribution introduced in Theorem 3.2 with constant $c$ and $\lambda = 1$. Let $A_{c,k} = \frac{c^2(1 - e^{-c} - \frac{c}{2})^{k-1}}{(1 - e^{-c})^k}$, where $k$ is a fixed predecided sampling size. Then, running DUET over $f$ using the LCB acquisition function found in (Chowdhury & Gopalan, 2017) at each BO iteration $t = 1, \ldots, T$ yields the following **attained average regret** (Chen et al., 2024b) upper bound with probability at least $1 - \delta$:*

$$\lim_{T \to \infty} \frac{\tilde{R}_T}{T} \leq \frac{6(\sqrt[4]{\delta} + \sqrt{k})}{\sqrt[4]{\delta}k} + 2A_{c,k} + \frac{\sqrt{2A_{c,k}}}{\sqrt[4]{\delta}}.$$

The proof is provided in App. B.3 and bounds $|f(r^*) - f(r_t)|$ and $\epsilon_t$ independently using BO regret analysis (Chen et al., 2024b; Chowdhury & Gopalan, 2017) and the error distribution defined in Theorem 3.2. Our Theorem's average regret indicates how close our algorithm converges to the optimal evaluation task loss with increasing BO iteration $T$ and different choices of sampling size $k$. Notice that because $c$ characterizes the error of our estimator in Theorem 3.2, a larger $c$ would decrease $A_{c,k}$ and our average regret. In addition, a larger sampling size $k$ reduces the estimation error of the inner problem (Theorem. 3.2), decreasing $A_{c,k}$ and reducing our regret bound, although our experiments (Sec. 6.2) show that setting $k = 1$ is sufficient to achieve good performance.

## 5 PRACTICAL CONSIDERATIONS

We are free to use any data selection methods in DUET's inner loop. We specifically highlighted IF as a data selection method because in our experiments, IF worked slightly better when paired with BO (see Fig. 6 for detailed ablation) as compared to other selection methods. It also has some interpretable advantages (Sec. A.6). Even though computing IF scores could be budget-intensive, practical tricks, such as parallel computation, Hessian approximation (Agarwal et al., 2017), pre-computation, or a smaller surrogate model can speed up computation.

If scaling to large-scale datasets is too compute-intensive, one could also use cheaper data selection methods in DUET, such as LESS (Xia et al., 2024) or TracIn (Pruthi et al., 2020) with some performance-tradeoff. In the extreme case, one can even resort to the uniform random estimator introduced in Sec. 3.2, which does not perform any data selection. We experimented with DUET paired with different data selection methods in Sec. 6.3 and discussed their actual compute-time in App. C.1. In addition, DUET's iterative optimization process is a feature: subjecting LLMs to multiple rounds of training in a feedback loop is a natural part of its deployment life-cycle.

## 6 EXPERIMENTS AND DISCUSSION

We conduct extensive experiments to showcase the effectiveness of DUET compared to other baselines. We optimize data mixtures with different methods based on multiple rounds of evaluation task performance feedback. Then, we fine-tune an LLM with the optimized data mixture. Lastly, we deploy the LLM on the evaluation task to evaluate how well the model has performed. We provide more details of our experimental setup and our algorithm computational cost in App. C.1.

### 6.1 EXPERIMENTAL SETUP

Our experiments are carried out by performing PEFT (Hu et al., 2021) of `Llama-3-8b-Instruct` (Touvron et al., 2023) across different LLM knowledge domains. We also ran our experiments with `Qwen2.5-7B-Instruct` (Qwen et al., 2025) and present the results in App. C.3. Our findings were similar even for different LLMs. The training data domains for LLM evaluation consists of 9 topics: **Wikitext** (Merity et al., 2016), **gsm8k** (Cobbe et al., 2021), **PubmedQA** (Jin et al., 2019), **HeadQA** (Vilares & Gómez-Rodríguez, 2019) , **SciQ** (Welbl et al., 2017), **TriviaQA** (Joshi et al., 2017), **TruthfulQA** (Lin et al., 2022), **Hellaswag** (Zellers et al., 2019), and **CommonsenseQA** (Talmor et al., 2019). We also varied the difficulty of the unseen task by making them out-of-domain (see captions of Fig. 4). Our LLM performance might have slight differences from existing papers, most likely due to evaluation setup differences, which we elaborate in App. C.1. For DUET, we "warm-started" the BO processes by evaluating 50 different random data mixtures and updated the GP with the performance observations .

We ran several baselines: **DoReMi** (Xie et al., 2023a) is a data-mixing approach that optimizes the data mixture in a distributionally robust manner. **LESS** (Xia et al., 2024) is a data-selection method based on data gradient similarities. The **Uniform weights** baseline uses a data mixture of uniform ratio across different domains. We ran our baselines for the same number of iterations as DUET and take the best performing result to ensure similar compute comparison. We also used DUET with a few different data selection methods: **DUET-IF** uses our IF-driven estimator (Eq. 5) to select data mixtures at each BO iteration; **DUET-UR**, introduced in Sec. 3.2, uses the uniform random estimator and randomly selects data mixtures that satisfy the proposed mixing ratio; **DUET-RH** (**R**emove **H**armful) removes 20% of data points with the lowest IF scores from each data domain, before performing sampling. **DUET-LESS** (Xia et al., 2024) and **DUET-logdet** (Wang et al., 2024b), which incorporate different data selection methods into DUET, were also used in our ablation studies (Fig. 6). We used a sampling size of $k = 1$ and BO iterations $T = 10$. We also constrained the total number of selected data points to $M = 10000$ with a temperature of $0.75$ in our LLMs. This makes the "feedback" (performance) of all valuation tasks *noisy*, similar to real-world tasks.

We also compared DUET with other baselines, such as **Aioli** (Chen et al., 2024a), **Multi-fidelity BO** (Yen et al., 2025), **online data-mixing** (Albalak et al., 2023a), alongside naive approaches: e.g., using more training tokens, random search or only data selection. Due to space constraints, we show these results in Table. 2. In general, DUET still finds better data mixtures than these baselines.

### 6.2 MAIN RESULT

**DUET finds more optimal data mixtures.** Our result (Fig. 4) shows that DUET finds better data mixtures within a few iterations of feedback loops. The first column in Fig. 4 consists of a relatively easier task where the evaluation domain is part of the training task domains. In this case, DUET (green plot) uses feedback from the evaluation task to find the optimal data mixture with more weights on the relevant training data domain, TruthfulQA. On the other hand, we observe the weakness of conventional methods which cannot exploit coarse feedback: DoReMi (orange dotted line) and LESS

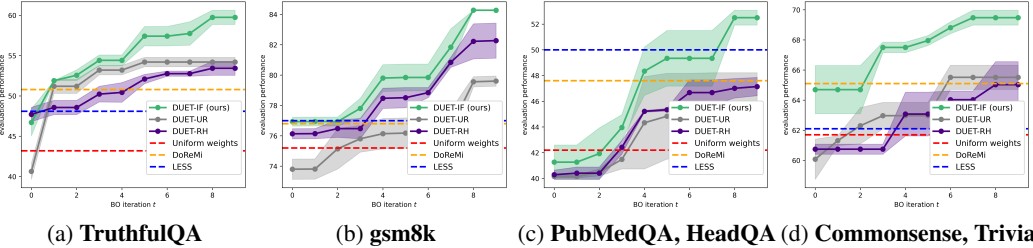

(a) **TruthfulQA**  (b) **gsm8k**  (c) **PubMedQA, HeadQA** (d) **Commonsense, Trivia**

Figure 4: Results on unseen LLM evaluation task domains over 10 iterations (higher is better) for `Llama-3-8b-Instruct`. Experiments were repeated with `Qwen2.5-7b-Instruct` in App. C.3. The caption shows the evaluation task. **Underlined evaluation tasks are harder** because the evaluation task domains are removed from the training data (i.e., out-of-domain). Results for more baselines are presented in Table. 1.

(orange dotted line) both cannot specifically adapt to the evaluation task and hence do not perform as well. In the 2nd, 3rd and 4th columns, we increased the difficulty of our evaluation task by removing the evaluation task domain from our training domains (**the evaluation task is now out-of-domain**). Surprisingly, DUET still can use feedback from the unseen task to automatically optimize the data mixture, achieving better LLM performance than other baselines. This suggests data from another training domain is still useful for the out-of-domain evaluation task (e.g., **Wikitext** data can still be helpful for mathematical questions in **gsm8k**). Hence, DUET is effective in both in-domain and out-of-domain tasks. In App. C.4, we qualitatively discuss the optimal mixing ratios found by DUET.

## 6.3 ABLATION EXPERIMENTS

While we have shown that DUET outperforms existing baselines, we also ran several ablations (using Fig. 4d) setting) to tease apart several components in DUET.

**Ablation of different components in DUET**. Fig. 5 shows the importance of both BO and data selection techniques in DUET. If we used a uniform data mixture to train an LLM, we can only achieve a baseline performance given by the red dotted line. With just BO, DUET automatically reconfigures the mixing ratio and attains performance gain (**A**). Next, by incorporating data selection methods, such as using IF in DUET-IF, we attain further performance gains (**B**) indicated by the green plot. Different data selection methods used in DUET also improves the LLM's performance to a different extent (**C**). Therefore, this affirms the importance of interleaving data selection and BO.

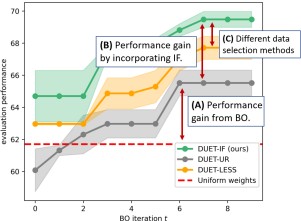

Figure 5: Ablation of different components of DUET.

**Ablation of using different data selection methods in DUET.** How do different data selection methods fare when used in DUET's inner loop? In Fig. 6, we found that IF outperforms other data selection methods (LESS, RH, log-det (Wang et al., 2024b)) when used in DUET's inner loop. This suggests that IF retrieves higher-quality (or remove lower-quality) data points at each iteration better than other methods. This aligns with our discussion in App. A.5 where we explained how IF, being able to remove low-quality data, yields better training data mixture in our unseen task setting. All in all, we are free to use different data selection techniques (each with different computational cost, performance) in DUET's inner loop.

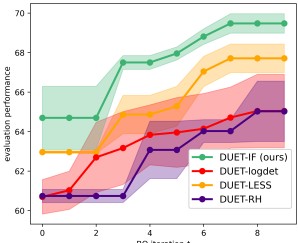

Figure 6: Ablation of using different data selection methods in DUET.

**Ablation of varying sampling size $k$.** Lastly, we also found that increasing sampling size $k$ in DUET's inner loop (Fig. 7) helps DUET find more optimal training data mixtures. This aligns with our theoretical findings from Theorem 4.1, which shows that larger $k$ improves DUET's convergence. In practical settings, if budget permits, LLM owners can fine-tune multiple copies of LLMs (i.e., increase $k$) to

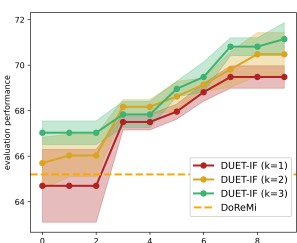

Figure 7: Ablation of sampling size $k$ in DUET.

improve DUET's performance. However, our results in Fig. 4 showed that even with $k = 1$, DUET outperforms other baselines.

## 7 CONCLUSION AND LIMITATIONS

Our paper proposes DUET, a novel algorithm that exploits multiple rounds of coarse, noisy feedback from a downstream unseen evaluation task to automatically optimize training data mixture for LLMs. Our approach offers an effective solution to address the unseen task setting, where fine-grained data information is unavailable (and conventional approaches fail). It is also quite flexible, allowing us to choose amongst different data selection methods in its inner loop. We provide theoretical guarantees of DUET and empirically show that it optimizes data mixtures in a variety of LLM evaluation tasks better than other baselines. One limitation is that our paper focused on LLM fine-tuning, but broadly speaking, we believe that DUET can be adapted to and would work equally well for pre-training. This leaves room for fruitful future research.

### ACKNOWLEDGMENTS

This research is supported by the National Research Foundation, Singapore under its National Large Language Models Funding Initiative (AISG Award No: AISG-NMLP-2024-001). In addition, this research is supported by the National Research Foundation, Singapore under its AI Singapore Programme (AISG Award No: AISG2-PhD/2023-01-039J). This research is part of the programme DesCartes and is supported by the National Research Foundation, Prime Minister's Office, Singapore under its Campus for Research Excellence and Technological Enterprise (CREATE) programme. Zhiliang Chen is supported by the Agency for Science, Technology and Research (A\*STAR), Singapore.

## 8 ETHICS STATEMENT

Our work strives to improve the performance of LLMs for the greater good. We do not foresee any ethical concerns related to our work. From our theoretical findings and experiments, our work can also handle noisy real-world feedbacks robustly.

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

## A    SUPPLEMENTARY MATERIAL

### A.1    REAL-WORLD EXAMPLES OF OUR PROBLEM SETTING

In our problem setting, (a) there is no direct access to the data (e.g., its domain, distribution, or labels) involved in the unseen evaluation task but (b) multiple rounds of coarse feedback (details covered in Sec. 2.2) can be gathered from the task using a trained LLM. Here, we provide several real-world examples in which such a setting occurs.

**End-to-end encrypted conversations between LLM and users.** This setting is specific to the conversational setting between a trained LLM and human users. LLM owners are interested in fine-tuning an LLM to converse well with some human-user demographics but due to real-world privacy concerns (Li et al., 2024), conversations between a deployed LLM and users are end-to-end encrypted during test-time (`openai.com/enterprise-privacy`). So, an LLM owner does not have any knowledge of the conversation domain or the (unlabeled or labeled) data seen during test-time. Instead, they only receive a feedback on how well the LLM has performed in the conversation (e.g., ratings from the human user, how long each user stays on the applicaton). The LLM owner can collect multiple rounds of feedback over a period of time. Hence, they can exploit this feedback to iteratively refine the training data mixture. Many chat-driven applications (e.g., whatsapp, telegram) nowadays use end-to-end encrypted chats, so our problem setting is relevant here.

**Model marketplace.** In addition, there are other scenarios in which a model owner needs to improve an ML model without having access to the data involved in the unseen evaluation task. For instance, an ML model owner might rent or sell an ML model in a model marketplace (e.g., `https://aws.amazon.com/marketplace/solutions/machine-learning`). However, the consumer might give feedback (e.g., how often the model makes mistakes) to the ML model owner in hope that the ML model owner can improve the model's performance on its own evaluation task. Furthermore, the data used by the consumer in its evaluation task are considered sensitive data, so the ML model owner does not know any data involved in the unseen evaluation task. Hence, the ML owner can only rely on feedback from the consumer to improve the training data mixture.

### A.2    MORE RELATED WORKS ON DATA MIXING AND SELECTION

Recently, a large class of data selection methods utilizing coresets (Zhang et al., 2024), diversity measures (Wang et al., 2024b), gradient information (Xia et al., 2024) or influence function (Koh & Liang, 2017) has been introduced to retrieve a smaller subset of data from an existing dataset. These data selection methods have become popular because they reduce training dataset size (which is an attractive feature when traning LLMs) and prior work (Xia et al., 2024) showed that training a model with strategically selected data points allows it to perform better. In addition, data mixing works (Xie et al., 2023a; Ge et al., 2025; Albalak et al., 2023b) have studied how to reweigh different data domains to produce optimal data mixtures, using distributionally robust optimization or entropy-based signals. However, these works, when used in isolation, **do not work well in our setting because they do not exploit feedback from an unseen evaluation task**. For example, even if we can retrieve a high-quality data subset from the training data domain, this domain might not even be relevant to the unseen evaluation task. Hence, data mixing and selection methods on their own **are not applicable to our setting** because they have no way to discern how relevant the training data domain is to the unseen task. Instead, our paper's algorithm interleaves BO and data selection method together to exploit feedback from the unseen evaluation task to optimize our training data mixture. Indeed, our experimental results in Sec. 6.2 show that DUET performs better than other data mixing and selection works.

### A.3    EXTENSIONS AND DISCUSSION OF DUET IN OTHER SPECIAL SETTINGS

Here, we discuss some extensions of DUET to other settings that fall beyond the scope of our paper. However, we find them insightful and useful when implementing DUET in practice.

**Should we re-fine-tune/re-train the LLM from scratch each time in DUET or continue training the model from the previous iteration?** Our problem formulation in Sec. 2.2 and theoretical findings (Sec. 4) assumes DUET re-train the LLM from the same initial checkpoint at every iteration. This is necessary to ensure our surrogate function landscape in GP remains consistent throughout the BO

process, allowing DUET to converge. From a practical perspective, we speculate that DUET will be less effective if we continue training an LLM from the previous iteration. This is because training data mixtures from earlier iterations might not be useful for the unseen evaluation task and the model might memorize (Tirumala et al., 2022) irrelevant information that are difficult to be overwritten in later BO iterations.

**DUET for extremely large datasets used in pre-training.** We can amortize the computational cost of IF computation by pre-computing and storing them beforehand (App. A.4) in our paper's fine-tuning setting. However, the size of datasets used in pre-training could be extremely large, which might still lead to large computational cost when computing IF scores of every data point in such datasets. To make computation faster, we can adopt methods in (Koh & Liang, 2017) to approximate hessian inversions when computing IF scores. We can also sample a smaller subset of data to compute IF scores, before training a neural network (Jethani et al., 2022) to predict the IF scores of other data points.

**Noisy feedback setting.** In some practical settings, the feedback from the unseen task is noisy. For instance, user ratings have a variance even within the same user demographics. How does DUET fare when the feedback from the unseen evaluation task is noisy? Fortunately, DUET is equally effective even when feedback is noisy. Feedback noise becomes part of the observation noise (Sec. 3.3) under the BO framework in DUET. In our experiments (Sec. 6.2), the evaluation task feedback is inherently noisy since LLM responses are probabilistic in nature, but DUET still performs well empirically.

### A.4 INFLUENCE FUNCTION AND ITS CALCULATIONS

Influence function (IF) (Koh & Liang, 2017) has been developed to study the influence of a single data point on an ML model's predictions. In this section we provide a summary of IF and its derivation. The influence of a data point $z$ on the loss of a test data point (or a set of test data points) $z_{\text{test}}$ for an ML model parameterized by $\theta$ is given by the closed-form expression:

$$\text{IF}_{z, z_{\text{test}}} = -\nabla_\theta L(z_{\text{test}}, \theta)^T H_\theta^{-1} \nabla_\theta L(z, \theta), \tag{6}$$

where $L$ is the loss function of the ML model and $H$ is the hessian of the ML model w.r.t. parameters $\theta$. In short, a data point is deemed more "influential" in reducing the model loss on a test data point if it has a higher IF score. As such, IF scores have also become a popular method in selecting data points which are more helpful in training an ML model.

In our work, we segregated a validation dataset from each data domain's dataset, in which we use to derive the IF score of every training data point in that domain w.r.t. the validation dataset (after fine-tuning an LLM over the training data till convergence). Then, we normalize these IF scores (for data points in each data domain), allowing us to perform weighted random sampling at every BO iteration of our algorithm, obtaining a data subset of size $n$ for a given data domain. This IF-weighted sampling is repeated for every data domain until we sample a dataset fulfilling the proposed mixing ratio at every BO iteration. Hence, the resulting data mixture contains more proportion of high-quality data points (based on IF scores). A summary of the IF-weighted sampling process for one data domain is given in Alg. 2. In our algorithm, we repeat this procedure for every data domain.

---

**Algorithm 2** IF-weighted sampling for **one data domain containing** dataset $D$

---

1: **Input:** number of data points $n$ required for the given data domain (taken from the mixing ratio proposed at current iteration). Dataset $D = \{x_1, x_2, ..., x_{|D|}\}$, Influence value of each data point in data domain dataset $D$: $\mathcal{I} \triangleq [I_1, I_2, \ldots, I_{|D|}]$, small constant $\epsilon$ to avoid degenerate-case normalization.

2: Normalize the IF scores into probabilities: $\mathcal{I}_{\text{normalized}} \triangleq [\frac{I_1 + \min(\mathcal{I}) + \epsilon}{\sum \mathcal{I}}, \frac{I_2 + \min(\mathcal{I}) + \epsilon}{\sum \mathcal{I}}, \ldots, \frac{I_{|D|} + \min(\mathcal{I}) + \epsilon}{\sum \mathcal{I}}]$

3: Perform weighed sampling from dataset $D$ according to weights given by $\mathcal{I}_{\text{normalized}}$ $n$ times.

---

**IF scores can be pre-computed and stored**. In addition, we just need to pre-compute the IF scores of every data point once before reusing them repeatedly at every BO iteration to perform IF-weighted sampling. This greatly improves our algorithm's efficiency and runtime, as compared to

other methods (see next section) which requires us to perform expensive computation every iteration. We provide the computation runtime of calculating IF scores in App. C.1.

### A.5 Discussion of using other data selection methods to solve inner optimization problem in DUET

Data selection methods (Albalak et al., 2024; Guo et al., 2024; Wang et al., 2024b) have been used to retrieve a representative subset of data from larger datasets. We note that in our work different data selection methods can be interchanged to produce different estimators for the inner problem solution in line 4 and 5 of Algorithm 1. For example, instead of using the IF-driven estimator which performs weighted sampling based on each data point's IF scores, one could use LESS (Xia et al., 2024) to retrieve data subsets for the inner optimization problem. However, our experiments (Fig. 6) have shown that other data selection methods perform slightly worse than IF when used in DUET's inner loop. We speculate that this occurs for a few reasons.

Specifically, IF (Koh & Liang, 2017) is effective at identifying non-useful data (e.g., nonsensical text, text with lots of spelling mistakes, blur images) and so IF will down-weigh low-quality datapoints when we sample from that data domain. Doing so is effective in DUET's setting because these nonsensical training data are unlikely to be useful for any tasks, so their removal can boost the performance of the selected data mixture on the unseen task. While LESS contains a similar formulation as IF, it merely consider the gradient dot-products and ignores the hessian of the loss function (Koh & Liang, 2017) during computation. Hence, it does not contain as much information as IF.

In addition, diversity-driven methods (Wang et al., 2024b; Zhang et al., 2024) tend to select training data subsets that are "most representative" of the training data domain. However, from observation, they tend to keep nonsensical data points in the final data mixture, which is not as effective as IF, which down-samples these points. Also, representative data of a training domain might not be useful for an unseen task if the task is not related. Lastly, when calculating the data log-determinant, we need to project data into embedding space with an embedding model, and hence the effectiveness of the embedding model also affects the selection process. Effectiveness aside, diversity-driven methods are also dependent on the mixing ratio chosen at each iteration. Therefore, we need to recompute the log-determinant (Wang et al., 2024b) or coreset (Zhang et al., 2024) at every iteration. On the contrary, IF scores can be pre-computed and stored prior to running DUET.

Therefore, different selection methods have different properties (above), but conceptually and empirically, we found IF to work better in our unseen task setting. Our ablation studies (Fig. 6) affirms our claim: using IF in DUET attains the highest performance as compared to selection methods such as LESS and log-determinant. We hope DUET can serve as a testbed for more advanced data-selection methods in the future.

### A.6 More discussion on the IF-driven estimator

Here, we provide more justification behind our choice of the IF-driven estimator.

**Why not take the data with the top-N IF scores instead of sampling?** One obvious alternative is to pick the data subset with the top IF scores (satisfying the given data ratio) or remove the datasubset with the lowest IF scores. We did not find this selection method effective in practice. Because IF-values are pre-computed and independent (App. A.4), we end up selecting the same few datapoints with top IF scores at every BO iteration in DUET. With the IF-weighted sampling, we select more diverse data points, yielding better data mixtures. This becomes more apparent when many data points have high IF scores and sampling provides us access to these data points at every iteration. Empirically, we also found that the deterministic approach performs worse (See DUET-RH in Sec. 6.2) than IF-weighted sampling.

Performing weighted-sampling with IF-scores not only retains the benefits of using IF-scores (we upweigh higher quality data while still having access to data points with moderate IF-scores), but also allows us to exploit additional computational resources to reduce the estimation error by increasing the sampling size. For instance, Theorem 3.2 shows that higher sampling size reduces the inner problem estimation variance and bias. Intuitively, an LLM owner could exploit more compute to

increase sampling size $k$ and sample more data mixtures and reduce the estimation error at every BO iteration. These estimation error variances are also handled gracefully in the BO framework.

In our experiments, we demonstrate that even with limited resource to sample and train an LLM once ($k = 1$), DUET still outperforms other baselines in our setting. In fact, Our ablation (Fig. 7) shows that increasing $k$ results in a performance boost, showcasing the benefits of sampling in real-world settings (where computational resources are available to make multiple queries each BO step).

### A.7 EMPIRICAL DISTRIBUTIONS OF ESTIMATORS FROM DIFFERENT DATA SELECTION METHODS

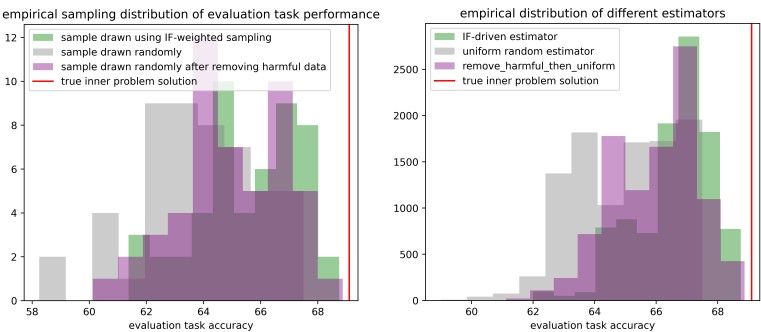

(a) **Empirical sampling distribution of** (b) **Empirical estimator distribution**
$\mathcal{L}_{\text{eval}}(\theta_\mathcal{X})$

Figure 8: (a): Empirical distribution of evaluation task accuracy $\mathcal{L}_{\text{eval}}(\theta_\mathcal{X})$ from each data mixture sample $\mathcal{X}$ (b): empirical distribution of the estimators introduced in Sec. 3.2. The green histogram is our method of performing IF-weighted sampling to obtain data mixtures. The gray histogram is simply randomly sampling data mixtures with no data selection methods. The purple histogram is the method of removing 20% of the data points with the lowest IF scores.

We have introduced the IF-driven estimator in Sec. 3.2 to estimate the solution of the inner problem. The IF-driven estimator performs IF-weighted sampling on data points from each data domain to produce data mixture samples (Eq. 5) constrained to a data mixing ratio $r$. Each data mixture sample is then used to train/fine-tune an LLM before obtaining a feedback on how well it has performed on the unseen evaluation task. Hence, this feedback based on each data mixture sample is also a sampling distribution that we can empirically observe. Fig. 8a shows the sampling distribution of the evaluation task performance obtained from each data mixture. Empirically, we see that the negative of this distribution is similar to a truncate exponential distribution mentioned in Theorem 3.2 (We consider the negative of this random variable because our paper considers the evaluation task loss, but empirically we maximize the evaluation task accuracy). In addition, the truncated exponential distribution is appropriate because it implies the unseen evaluation task loss is upper bounded at $y_r^* + c$ for a non-negative constant $c$; this is a reasonable assumption because many real-world feedbacks are bounded (e.g. user ratings).

We also plot the empirical distribution of the IF-driven estimator introduced in Eq. 5 in Fig. 8b. The distribution coincides with the estimator's distribution (formally, $y_r^* + \epsilon$) introduced in Theorem 3.2. From the estimator's distribution, we see that the IF-driven estimator (green histogram) has the lower bias and variance as compared to other estimators.

## B PROOFS

### B.1 PROOF OF THEOREM 3.1

**Theorem 3.1.** $\mathcal{X}^*$, *the optimal set of data points from* $\mathcal{D}$, *is the solution of the original problem* 2 *iff* $r^* = ratio(\mathcal{X}^*)$ *is the optimal mixing ratio solution of the reparameterized problem:*

$$\min_{r \in \mathbb{R}^n} \min_{\mathcal{X} \in S_r} \quad \mathcal{L}_{eval}(\theta_\mathcal{X}), \tag{3}$$

where $S_r \triangleq \{\mathcal{X} : \mathcal{X} \in \mathcal{D}, ratio(\mathcal{X}) = r, |\mathcal{X}| = M\}$ and $ratio(\mathcal{X}) = r$ means that the data points in $\mathcal{X}$ satisfies the mixing ratio $r \in \mathbb{R}^N$ from $n$ data domains and $\|r\|_1 = 1$.

*Proof.* Theorem. 3.1 can be proven in two steps. First, we restate the theoretical results from (Chen et al., 2024b) in Lemma B.1. This Lemma reparameterizes any optimization problem ($\min_x f(x)$) (while retaining the solution set *exactly*) under some regular assumptions:

**Lemma B.1.** *Let $x \in \mathbb{R}^d$ and $y \in \mathbb{R}^n$. Also, consider well-defined functions $f$ over $\mathbb{R}^d \to \mathbb{R}$ and $g$ over $\mathbb{R}^d \to \mathbb{R}^n$. Then $x^*$ is a solution of $\arg\min_x f(x)$ if and only if $y^* = g(x^*)$ is a solution of the second optimization problem over domain $\{y \mid \exists x, g(x) = y\}$ :*

$$\min_y \min_x \quad f(x)$$
$$s.t. \quad g(x) = y$$

The proof of Lemma B.1 can be found in Lemma C.1 of (Chen et al., 2024b). Next, we show that the objective function of problem 3 introduced in our optimization problem satisfies these assumptions, allowing us to apply the Lemma B.1 directly.

In our setting, we set $x \triangleq \mathcal{X}$, $f(x) \triangleq \mathcal{L}_{\text{eval}}(\theta_{\mathcal{X}})$ and $g(x) \triangleq ratio(\mathcal{X})$. We can see that both functions are well-defined, where for any chosen input $\mathcal{X}$, there certainly exists an observed evaluation task loss $\mathcal{L}_{\text{eval}}(\theta_{\mathcal{X}})$ and mixing ratio $ratio(\mathcal{X})$. Lastly, by setting $y \triangleq r$, our optimization problem in problem 3 is of the identical form of the optimization problem shown in Lemma B.1. Therefore, our reparameterization process is valid.

$\square$

## B.2 PROOF OF THEOREM 3.2

**Theorem 3.2.** *Let $\{\mathcal{X}_1^{IF}, \ldots, \mathcal{X}_k^{IF}\}$ be $k$ data mixture samples drawn from $S_r$ using IF-weighted sampling. Furthermore, assume each independent sample $\mathcal{L}_{eval}(\theta_{\mathcal{X}_i^{IF}})$ follows the shifted truncated exponential distribution $y_r^* + \exp_t(\lambda, c)$, for $i = 1, 2, \ldots, k$ where $\exp_t(\lambda, c)$ is a truncated exponential distribution governed by rate parameter $\lambda$ and truncated at $c > 0$. Then, the IF-driven estimator $\widetilde{y_r^*}$ defined in Eq. 5 is a random variable: $y_r^* + \epsilon$, where $y_r^*$ is the true inner problem solution of Eq. 4 and $\epsilon$ is a random noise variable with probability density function (PDF):*

$$PDF_\epsilon(u) = \frac{\lambda k e^{-\lambda u}}{1 - e^{-\lambda c}} \left(\frac{e^{-\lambda u} - e^{-\lambda c}}{1 - e^{-\lambda c}}\right)^{k-1} \quad on \ u \in [0, c] \ .$$

*Proof.* Let $X_1, X_2, \ldots, X_k$ be $k$ samples randomly drawn from a sampling distribution and $X_{\min} = \min\{X_1, X_2, \ldots, X_k\}$. This scenario mirrors the setting in Theorem 3.2. Our goal is to derive the distribution of $X_{\min}$ and show that it is exactly the same as the distribution of $\widetilde{y_r^*}$ shown in the Theorem 3.2.

If each random sample $X_i \sim \exp_t(\lambda, c)$, we first use a commonly known result (Chen et al., 2024b) that the CDF of any truncated distribution on $[0, c]$ is $\frac{F(u) - F(0)}{F(c) - F(0)}$ where $F$ is the CDF of the original distribution. Also, we note that for the untruncated exponential distribution, $F(u) = 1 - e^{-\lambda u}$. Hence, The CDF of $X_{\min}$ is

$$\begin{aligned}
\text{cdf}_{(X_{\min})}(u) &= 1 - \mathbb{P}(X_{\min} \geq u) \\
&= 1 - \mathbb{P}(X_1 \geq u, X_2 \geq u, \ldots, X_k \geq u) \\
&= 1 - \left(1 - \frac{1 - e^{-\lambda u}}{1 - e^{-\lambda c}}\right)^k, \quad 0 \leq u \leq c.
\end{aligned}$$

and so the PDF of $X_{\min}$ can be computed as

$$\begin{aligned}
\text{PDF}_{(X_{\min})}(u) &= \frac{\partial}{\partial u} F_{(X_{\min})}(u) \\
&= \frac{\lambda k e^{-\lambda u}}{1 - e^{-\lambda c}} \left( \frac{e^{-\lambda u} - e^{-\lambda c}}{1 - e^{-\lambda c}} \right)^{k-1}, \quad 0 \le u \le c.
\end{aligned}$$

In the original Theorem, each sample $X_i$ follows the shifted truncated exponential distribution $y_r^* + \exp_t(\lambda, c)$ where $y_r^*$ is a constant. Hence, we can see that our estimator has the distribution of $y_r^* + X_{\min}$ where $X_{\min}$ has the PDF above. Hence, the Theorem is proven by setting the random variable $\epsilon = X_{\min}$.

$\square$

### B.3 PROOF OF THEOREM 4.1

**Theorem 4.1.** *Let $f$ be the outer problem objective defined in Sec. 3.3 with bounded RKHS norm: $||f||_\kappa = \sqrt{\langle f, f \rangle_\kappa}$. Also, let our IF-driven estimator for the inner problem solution be governed by the error distribution introduced in Theorem 3.2 with constant $c$ and $\lambda = 1$. Let $A_{c,k} = \frac{c^2 (1 - e^{-c} - \frac{c}{2})^{k-1}}{(1 - e^{-c})^k}$, where $k$ is a fixed predecided sampling size. Then, running DUET over $f$ using the LCB acquisition function found in (Chowdhury & Gopalan, 2017) at each BO iteration $t = 1, \ldots, T$ yields the following **attained average regret** (Chen et al., 2024b) upper bound with probability at least $1 - \delta$:*

$$\lim_{T \to \infty} \frac{\tilde{R}_T}{T} \le \frac{6(\sqrt[4]{\delta} + \sqrt{k})}{\sqrt[4]{\delta} k} + 2 A_{c,k} + \frac{\sqrt{2 A_{c,k}}}{\sqrt[4]{\delta}}.$$

*Proof.* We provide the proof of the sub-linear $\tilde{R}_T$ growth of DUET in Theorem 4.1 by establishing upper bounds of $|\mu_t(x) - f(x)|$ and $\epsilon_t$ separately at each BO iteration $t$ and use the independence rule to bound their sum. To do so, we introduce the following two Lemmas.

Our first Lemma is taken from from known literature on Kernelized Bandits (Chowdhury & Gopalan, 2017) and provides the upper bound on difference between $f(x_t)$ and $\mu_t(x)$ at each BO iteration $t$.

**Lemma B.2.** *Let $||f||_\kappa = \sqrt{\langle f, f \rangle_\kappa} \le B$. Also, assume that the observation noise associated with each BO iteration is $R$-sub-Gaussian with $R > 0$. Then with probability at least $1 - \delta$, the following holds for BO iteration $t \le T$:*

$$|\mu_t(x) - f(x)| \le \left( B + R \sqrt{2(\gamma_t + 1 + \ln(1/\delta))} \right) \sigma_t(x) \tag{7}$$

*where $\gamma_t$ is the maximum information gain after $t$ observations and $\mu_t(x), \sigma_t^2(x)$ are mean and variance of posteror distribution of GP defined in Equation 1, with $\lambda = 1 + 2/T$.*

Our second Lemma attempts to bound the expectation and variance of $\epsilon_t$, the non-negative observation noise (in our case, it corresponds to the estimation error involved in solving the inner problem) at each BO iteration $t$. These expectation and variance will be used later to bound our cumulative regret.

**Lemma B.3.** *Let each observation noise $\epsilon_t$ of BO iteration $t$ follow the same probability distribution as $\epsilon$ defined in Theorem 3.2 with sampling size $k$ probability density function $f_{\epsilon_t}(u) = \frac{\lambda k e^{-\lambda u}}{1 - e^{-\lambda c}} \left( \frac{e^{-\lambda u} - e^{-\lambda c}}{1 - e^{-\lambda c}} \right)^{k-1}$ with $0 < c \le 1$, $\lambda = 1$ and $u \in [0, c]$, then $\mathbb{E}(\epsilon_t) \le \frac{6}{k} + \frac{2c^2((1 - e^{-c}) - \frac{c}{2})^{k-1}}{(1 - e^{-c})^k}$ and $\text{Var}(\epsilon_t) \le \mathbb{E}(\epsilon_t)$.*

*Proof.* For $\lambda = 1$, we have that $f_{\epsilon_t}(u) = \frac{k e^{-u}}{1 - e^{-c}} \left( \frac{e^{-u} - e^{-c}}{1 - e^{-c}} \right)^{k-1}$ with $0 < c < 1$ and $u \in [0, c]$. Then, the expectation:

$$
\begin{aligned}
\mathbb{E}(\epsilon_t) &= \int_0^c u f_{\epsilon_t}(u)\, du \\
&= \int_0^c \frac{uke^{-u}}{1-e^{-c}} \left( \frac{e^{-u}-e^{-c}}{1-e^{-c}} \right)^{k-1} du \\
&= \frac{k}{(1-e^{-c})^k} \int_0^c ue^{-u} \left( e^{-u}-e^{-c} \right)^{k-1} du \\
&\overset{(1)}{\leq} \frac{k}{(1-e^{-c})^k} \int_0^c u \left( e^{-u}-e^{-c} \right)^{k-1} du \\
&\overset{(2)}{\leq} \frac{k}{(1-e^{-c})^k} \int_0^c u \left( \left(1-\frac{u}{2}\right)-e^{-c} \right)^{k-1} du \\
&\overset{(3)}{\leq} \frac{k}{(1-e^{-c})^k} \left( \frac{(u-2(1-e^{-c}))((1-e^{-c})-\frac{u}{2})^{k-1}(2(1-e^{-c})+(k-1)u+u)}{k(k+1)} \right) \Bigg|_{u=0}^{u=c} \\
&\overset{(4)}{=} \frac{1}{(1-e^{-c})^k} \left( \frac{(c-2(1-e^{-c}))((1-e^{-c})-\frac{c}{2})^{k-1}(2(1-e^{-c})+kc)+4(1-e^{-c})^{k+1}}{k+1} \right) \\
&\overset{(5)}{\leq} \frac{4(1-e^{-c})^{k+1}}{(k+1)(1-e^{-c})^k} + \frac{2kc^2((1-e^{-c})-\frac{c}{2})^{k-1}}{(k+1)(1-e^{-c})^k} + \frac{2((1-e^{-c})-\frac{c}{2})^{k-1}(1-e^{-c})}{(k+1)(1-e^{-c})^k} \\
&\overset{(6)}{\leq} \frac{6}{k} + \frac{2c^2((1-e^{-c})-\frac{c}{2})^{k-1}}{(1-e^{-c})^k}
\end{aligned}
$$
$$(8)$$

where $\overset{(1)}{\leq}$ makes use of the fact that $e^{-\lambda u} \leq 1$ for $u \in [0,c]$ with $c > 0$, $\overset{(2)}{\leq}$ uses the inequality $e^{-u} \leq 1-\frac{u}{2}$ for $u \in [0,c]$, and $c \leq 1$, $\overset{(3)}{=}$ uses the fact that $e^{-\lambda c} < 1$, $\overset{(4)}{=}$ is derived by solving the definite integral by parts and substitution and $\overset{(4)}{=}$ simplifies the upper bound with algebraic manipulation.

Next, the upper bound of the variance of $\epsilon_t$ can be derived by

$$
\begin{aligned}
\mathrm{Var}(\epsilon_t) &= \int_0^c u^2 f_{\epsilon_t}(u)\, du \\
&\overset{(1)}{\leq} c \int_0^c u f_{\epsilon_t}(u)\, du \\
&\overset{(2)}{\leq} \int_0^c u f_{\epsilon_t}(u)\, du \\
&= \mathbb{E}(\epsilon_t)
\end{aligned}
$$
$$(9)$$

where $\overset{(1)}{\leq}$ makes use of the fact that $\epsilon_t$ lies in $[0,c]$ and $\overset{(2)}{\leq}$ makes use of the fact that $0 < c \leq 1$. This completes the proof on the bounds on $\mathbb{E}(\epsilon_t)$ and $\mathrm{Var}(\epsilon_t)$. $\qquad \square$

Next, we observe that $x_t$ at each BO iteration $t$ is chosen via the IGP-LCB acquisition function (i.e., $x_t = \arg\min_x \mu_{t-1}(x) - \beta_t \sigma_{t-1}(x)$ and $\beta_t = B + R\sqrt{2(\gamma_{t-1}+1+\ln(1/\delta_1))}$ where the observation noise associated with each BO iteration is $R$-sub Gaussian). Thus, we can see that at each iteration $t \geq 1$, we have $-\mu_{t-1}(x_t) + \beta_t\sigma_{t-1}(x_t) \geq -\mu_{t-1}(x^*) + \beta_t\sigma_{t-1}(x^*)$. It then follows

that for all $t \geq 1$ and with probability at least $1 - \delta_1$,

$$
\begin{aligned}
|f(x_t) - f(x^*)| &\overset{(1)}{\leq} f(x_t) - \mu_{t-1}(x^*) - \beta_t \sigma_{t-1}(x^*) \\
&\overset{(2)}{\leq} f(x_t) - \mu_{t-1}(x_t) + \beta_t \sigma_{t-1}(x_t) \\
&\leq \beta_t \sigma_{t-1}(x_t) + |\mu_{t-1}(x_t) - f(x_t)| \\
&\leq 2\beta_t \sigma_{t-1}(x_t)
\end{aligned}
\tag{10}
$$

Therefore, by setting $\delta_1 = \delta_2 = \sqrt{\delta}$, it follows that with probability $1 - \delta$ (this follows by rule of independence applied to the upper bound of events $\sum_{t=1}^{T} |f(x_t) - f(x^*)|$ and $\sum_{t=1}^{T} \epsilon_t$) that our **attained cumulative regret** can be bounded as

$$
\begin{aligned}
\tilde{R}_T &= \sum_{t=1}^{T} |\tilde{y}_t - f(x^*)| \\
&= \sum_{t=1}^{T} |f(x_t) - f(x^*) + \epsilon_t| \\
&\overset{(1)}{=} \sum_{t=1}^{T} |f(x_t) - f(x^*)| + \sum_{t=1}^{T} \epsilon_t \\
&\overset{(2)}{\leq} 2\beta_T \sum_{t=1}^{T} \sigma_{t-1}(x_t) + \sum_{t=1}^{T} \epsilon_t \\
&\overset{(3)}{=} 2\left(B + R\sqrt{2(\gamma_T + 1 + \ln(1/\sqrt{\delta}))}\right) \sum_{t=1}^{T} \sigma_{t-1}(x_t) + \sum_{t=1}^{T} \epsilon_t \\
&\overset{(4)}{\leq} 2\left(B + R\sqrt{2(\gamma_T + 1 + \ln(1/\sqrt{\delta}))}\right) \sum_{t=1}^{T} \sigma_{t-1}(x_t) + \sum_{t=1}^{T} \mathbb{E}(\epsilon_t) + \sum_{t=1}^{T} \sqrt{\frac{\mathrm{Var}(\epsilon_t)}{\delta_2}} \\
&\overset{(5)}{=} 2\left(B + R\sqrt{2(\gamma_T + 1 + \ln(1/\sqrt{\delta}))}\right) O(\sqrt{T\gamma_T}) + \sum_{t=1}^{T} \mathbb{E}(\epsilon_t) + \sum_{t=1}^{T} \sqrt{\frac{\mathrm{Var}(\epsilon_t)}{\delta_2}} \\
&= O\left(\sqrt{T}(B\sqrt{\gamma_T} + R\gamma_T)\right) + \sum_{t=1}^{T} \mathbb{E}(\epsilon_t) + \sum_{t=1}^{T} \sqrt{\frac{\mathrm{Var}(\epsilon_t)}{\delta_2}} \\
&\overset{(6)}{=} O\left(\sqrt{T}(B\sqrt{\gamma_T} + \frac{c^2 \gamma_T}{4})\right) + \sum_{t=1}^{T} \mathbb{E}(\epsilon_t) + \sum_{t=1}^{T} \sqrt{\frac{\mathrm{Var}(\epsilon_t)}{\delta_2}}
\end{aligned}
\tag{11}
$$

where we have followed the attained cumulative regret proof in (Chen et al., 2024b) closely and used the following facts:

- $\overset{(1)}{=}$ uses the fact that $\epsilon_t$ is non-negative in our problem setting (Theorem. 3.2).

- $\overset{(2)}{\leq}$ is derived from Eq. equation 10.

- $\overset{(3)}{=}$ uses the definition of $\beta_T$ in IGP-LCB acquisition function (Chowdhury & Gopalan, 2017) w.r.t. $\delta_1 = \sqrt{\delta}$

- $\overset{(4)}{\leq}$ uses Chebyshev's inequality over $\epsilon_t$ with probability at least $1 - \delta_2$.

- $\overset{(5)}{=}$ uses $\sum_{t=1}^{T} \sigma_{t-1}(x_t) \leq O(\sqrt{T\gamma_T})$ as shown in **Lemma 4** by Chowdhury & Gopalan (Chowdhury & Gopalan, 2017).

- $\overset{(6)}{=}$ uses the fact that $\epsilon_t$ is bounded on $[0, c]$ and all bounded random variables are R-sub-Gaussian with $R = \frac{c^2}{4}$ (Arbel et al., 2019).

Next, we need to derive the upper bound of $\sum_{t=1}^{T} \mathbb{E}(\epsilon_t) + \sum_{t=1}^{T} \sqrt{\frac{\text{Var}(\epsilon_t)}{\delta_2}}$ w.r.t. $T$. This can be done by using the upper bound of the expectation and variance of $\epsilon_t$ proven in Lemma B.3:

$$\sum_{t=1}^{T} \mathbb{E}(\epsilon_t) + \sum_{t=1}^{T} \sqrt{\frac{\text{Var}(\epsilon_t)}{\delta_2}} \overset{(1)}{\leq} \sum_{t=1}^{T} \left( \frac{6}{k} + \frac{2c^2((1-e^{-c}) - \frac{c}{2})^{k-1}}{(1-e^{-c})^k} \right) + \sum_{t=1}^{T} \sqrt{\frac{6}{\delta_2 k} + \frac{2c^2((1-e^{-c}) - \frac{c}{2})^{k-1}}{\delta_2(1-e^{-c})^k}}$$

$$= \frac{6T}{k} + \frac{2Tc^2((1-e^{-c}) - \frac{c}{2})^{k-1}}{(1-e^{-c})^k} + T\sqrt{\frac{6}{\delta_2 k} + \frac{2c^2((1-e^{-c}) - \frac{c}{2})^{k-1}}{\delta_2(1-e^{-c})^k}}$$

$$(12)$$

where $\overset{(1)}{\leq}$ uses Lemma B.3 directly.

Then, it follows from Eq. 11 and 12 that with probability $1-\delta$ and $\delta_2 = \sqrt{\delta}$, the **attained cumulative regret** $\tilde{R}_T$ at iteration $T$ is upper bounded by:

$$\tilde{R}_T \leq O\left( \sqrt{T}(B\sqrt{\gamma_T} + \frac{c^2\gamma_T}{4}) \right) + \frac{6T}{k} + \frac{2Tc^2((1-e^{-c}) - \frac{c}{2})^{k-1}}{(1-e^{-c})^k} + T\sqrt{\frac{6}{\delta_2 k} + \frac{2c^2((1-e^{-c}) - \frac{c}{2})^{k-1}}{\delta_2(1-e^{-c})^k}}$$

$$(13)$$

Finally we set $A_{c,k} = \frac{c^2(1-e^{-c} - \frac{c}{2})^{k-1}}{(1-e^{-c})^k}$. As $T \to \infty$, with probability $1-\delta$ and $\delta_2 = \sqrt{\delta}$, the attained *average* regret converges to:

$$\lim_{T \to \infty} \frac{\tilde{R}_T}{T} \overset{(1)}{\leq} \frac{6}{k} + \frac{2((1-e^{-c}) - \frac{c}{2})^{k-1}}{(1-e^{-c})^k} + \sqrt{\frac{6}{\delta_2 k} + \frac{2((1-e^{-c}) - \frac{c}{2})^{k-1}}{\delta_2(1-e^{-c})^k}}$$

$$\overset{(2)}{\leq} \frac{6}{k} + \sqrt{\frac{6}{\delta_2 k}} + 2A_{c,k} + \sqrt{\frac{2A_{c,k}}{\delta_2}}$$

$$\leq \frac{6(\sqrt[4]{\delta} + \sqrt{k})}{\sqrt[4]{\delta}k} + 2A_{c,k} + \sqrt{\frac{2A_{c,k}}{\delta_2}}$$

$$(14)$$

where $\overset{(1)}{\leq}$ divides Eq. 13 by $T$ throughout, eliminating the $O$ expression and $\overset{(2)}{\leq}$ uses the subsitition of $A_{c,k}$ and triangle inequality. This completes our proof for the attained average regret in Theorem 4.1.

$\square$

## B.4 EXTENDING THEORETICAL ANALYSIS BASED ON DIFFERENT DATA SELECTION METHODS

Readers might be interested in how different data selection methods used to create different estimators affect our theoretical analysis. Here, we provide details on how one could replicate our paper's theoretical analysis to different estimators.

**Step 1. Establish the sampling distribution of $\mathcal{L}_{\text{eval}}(\theta_{\mathcal{X}})$.** Using a particular data selection method, one obtains $k$ data mixture samples $\{\mathcal{X}_1, \ldots, \mathcal{X}_k\}$ (in our paper, these samples are obtained via weighted sampling based on each data point's IF scores). Then, one trains an LLM for each data mixture and obtain the evaluation task loss for each resulting LLM, yielding $\{\mathcal{L}_{\text{eval}}(\theta_{\mathcal{X}_1}), \ldots, \mathcal{L}_{\text{eval}}\theta_{\mathcal{X}_k}\}$. From this set, one can empirically derive the sampling distribution of each sample $\mathcal{L}_{\text{eval}}(\theta_{\mathcal{X}_i})$. In Theorem 3.2, we assumed that each sample $\mathcal{L}_{\text{eval}}(\theta_{\mathcal{X}_i})$ follows the truncated exponential distribution. However, different data selection methods would certainly lead to different empirical sampling distributions.

**Step 2. Derive an estimator's empirical distribution**. Next, we need to theoretically derive the 1st-order statistic (Arnold et al., 2008) of the empirical sampling distribution from Step 1, since we use the 1st-order statistic as our estimator. The procedure to do so is shown in App. B.2 and uses a

fairly standard procedure to derive the distribution of order statistics. For subsequent analysis to be tractable, the PDF of the 1st-order statistic should have a closed form (hence, a simpler sampling distribution in Step 1 is preferred). More importantly, the estimator's empirical distribution **should be R-sub-gaussian** for a fixed $R > 0$. This is because for the regret-analysis proof in Eq. 11 to hold true, the observation noise in the BO process should be R-sub-Gaussian. Fortunately, a large family of random distributions, including our IF-driven estimator introduced in this paper, are all R-sub-Gaussian (e.g., exponential family, all bounded random variables).

**Step 3. Derive the upper bound of estimator's expectation and variance**. Next, we derive the upper bound of the 1st-order statistic's expectation and variance as shown in Lemma. B.3.

**Step 4. Derive attainable cumulative regret**. Lastly, we analyze the convergence rate of our algorithm using the growth of *attained cumulative regret* (Chen et al., 2024b) $\tilde{R}_T = \sum_{t=1}^{T} |\widetilde{y_{r_t}^*} - f(r_t)| = \sum_{t=1}^{T} |f(r^*) + \epsilon_t - f(r_t)|$ for $T$ BO iterations. Since the error term $\epsilon_t$ has the same expectation and variance of our estimator, we can use the results from Step 3 to derive our regret bound (as shown in Eq. 11).

## C  ADDITIONAL EXPERIMENTAL RESULTS AND DISCUSSIONS

### C.1  ADDITIONAL DETAILS ON EXPERIMENTAL SETUP

In this section, we provide additional details in our experiments for ease of reproduceability. Throughout our experiments, we used the SE kernel with lengthscale parameters learned from historical observations via maximum-likelihood (Williams & Rasmussen, 2006). In our LCB acquisition function (Greenhill et al., 2020), we set $\beta_t = 0.5$ (see Alg. 1) throughout our experiments. Furthermore, we need to perform constrained BO (Gardner et al., 2014) in our experiments because the inputs to our optimization problem is a data mixing ratio $r$ whose sum of entries is constrained to 1. BoTorch allows us to implement such constraints (`botorch.org/docs/constraints`) easily. All evaluation for language tasks is done on **llm-harness** (Gao et al., 2024) with default 3-shot settings with no chain-of-thought or special prompting techniques. Hence, it is possible some of our paper's results differ from those reported in other papers (due to different prompting and inference settings). However, our paper's emphasis is on improving the LLM's performance with a few rounds refinement on the training data mixture. Hence, we expect DUET to work well even in other inference settings. We treat the evaluation results on llm-harness as the feedback observed in our problem setting. For methods (e.g., uniform mixture, LESS, DoReMi) which do not use feedback, we repeat them 10 times to ensure fairness in comparison with DUET and show the best LLM performance in our results.

To train the LLM, we used a LoRA (Hu et al., 2021) rank of 128 and the Adam optimizer (Kingma & Ba, 2017) with initial learning rate of 1e-5. The specific hyperparameters surrounding our optimizers can be found in our code, which we have released. Each iteration of model fine-tuning is done in 1 epoch on a L40 Nvidia GPU and takes approximately an hour. So, performing 10 BO iterations take 10 hours to run. In reality, the optimization process (all 10 iterations) will be carried out over a long span of time (e.g., weeks, or even months) as part of the LLM deployment life-cycle. So this is a reasonable amount of compute time.

**IF computation**. To derive the IF scores of our training data, we remove 10% of the training data from each data domain and treat it as the validation set. Then, we fine-tune a separate LLM for each data domain (using the same setting as above and the same model type as that in our experiments), before deriving the IF score of every data point from each data domain based on the converged LLM and the validation dataset. Using 4 Nvidia L40 GPUS, we were able to compute the IF scores of TriviaQA (containing around 170k data points) in around 2-3 hours with the torch- influence library (`https://github.com/alstonlo/torch-influence`). Smaller datasets required even shorter computation time. Certainly, these runtimes are reasonable in practical settings, since we only need to compute the IF scores once and store them before running DUET.

**Other data selection methods.** We can also use other data selection methods in DUET's inner loop. For instance, LESS (Xia et al., 2024) and TracIn (Pruthi et al., 2020) uses around 4-5 hours to build a data-gradient store. On the other hand, diversity-driven selection techniques (Wang et al., 2024b)

are usually more computationally expensive, taking more than 30 hours to select the top 10000 data points.

## C.2 COMPARISON WITH OTHER BASELINES

One alternative baseline is to simply fine-tune the LLM on a training dataset for multiple more training tokens (and epoch) and compare it with DUET. In Table 1, we fine-tuned `Llama-3-8b-Instruct` for more epochs and training tokens on each training domain and evaluated on our evaluation task. The results show that DUET-IF attains better results because it can exploit the feedback from the task.

Table 1: Performance of models trained on different datasets across evaluation tasks.

| Train ↓ Eval. → | TruthfulQA | gsm8k | PubmedQA+HeadQA | Commonsense+TriviaQA |
|---|---|---|---|---|
| Wikitext | 42.8 | 70.4 | 40.6 | 59.9 |
| gsm8k | 47.2 | 86.1 | 43.3 | 64.1 |
| Pubmed | 43.3 | 71.5 | 49.3 | 58.4 |
| HeadQA | 45.0 | 75.2 | 50.2 | 60.0 |
| SciQ | 45.6 | 75.6 | 44.6 | 63.4 |
| TruthfulQA | 59.0 | 74.0 | 43.8 | 61.0 |
| Hellaswag | 46.1 | 72.1 | 43.2 | 60.4 |
| CommonsenseQA | 50.1 | 73.3 | 47.2 | 65.8 |
| TriviaQA | 51.2 | 70.1 | 48.1 | 66.5 |
| DUET-IF (ours) | **59.8** | **84.2** | **52.4** | **69.6** |

Next, we compared DUET (paired with different data selection methods) with a variety of naive baselines. **Aioli** (Chen et al., 2024a), **Multi-Fid** (Yen et al., 2025) are two baselines that use domain reweighting and multi-fidelity BO to optimize data mixtures. **IF** just picks the top $M = 20000$ datapoints with the highest influence scores. **Random** just selects a random subset of data, but we subjected it to more training epochs and $M = 50000$ data points to ensure equal compute comparison.

Table 2: Performance of other baselines across evaluation tasks.

| Other baselines | TruthfulQA | gsm8k | PQA+HQA | Commonsense, TriviaQA |
|---|---|---|---|---|
| Aioli (Chen et al., 2024a) | $51.1_{\pm 0.7}$ | $76.5_{\pm 1.2}$ | $48.8_{\pm 0.5}$ | $63.7_{\pm 1.0}$ |
| Multi-Fid (Yen et al., 2025) | $52.8_{\pm 0.9}$ | $73.9_{\pm 1.3}$ | $47.2_{\pm 0.6}$ | $65.2_{\pm 0.8}$ |
| ODM (Albalak et al., 2023a) | $46.1_{\pm 1.1}$ | $77.3_{\pm 0.4}$ | $45.8_{\pm 1.2}$ | $60.1_{\pm 0.7}$ |
| IF only | $50.8_{\pm 0.5}$ | $76.9_{\pm 0.8}$ | $47.7_{\pm 0.9}$ | $57.8_{\pm 0.6}$ |
| Random | $49.3_{\pm 0.8}$ | $64.3_{\pm 1.4}$ | $41.2_{\pm 0.7}$ | $57.3_{\pm 1.0}$ |
| Uniform + more training tokens | $51.6_{\pm 0.8}$ | $64.4_{\pm 1.3}$ | $44.5_{\pm 1.2}$ | $59.2_{\pm 1.6}$ |
| DUET-IF (ours) | $\mathbf{59.8}_{\pm 0.6}$ | $\mathbf{84.2}_{\pm 1.1}$ | $\mathbf{52.4}_{\pm 0.9}$ | $\mathbf{69.6}_{\pm 0.8}$ |
| DUET-LESS (ours) | $\mathbf{58.7}_{\pm 1.0}$ | $\mathbf{80.5}_{\pm 0.7}$ | $\mathbf{50.8}_{\pm 0.9}$ | $\mathbf{67.6}_{\pm 1.3}$ |

## C.3 ADDITIONAL EXPERIMENTAL RESULTS WITH QWEN2.5-7B-INSTRUCT

We repeated our experiments with `Qwen2.5-7B-Instruct` in Fig. 9 and observe that DUET still can optimize data mixtures better than other baselines. This indicates that the effectiveness of DUET is independent of the model choice. Hence, we expect DUET to work well for other models as well.

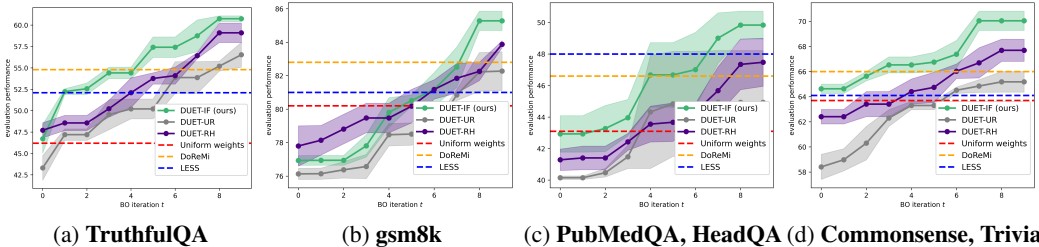

(a) **TruthfulQA**   (b) **gsm8k**   (c) **PubMedQA, HeadQA** (d) **Commonsense, Trivia**

Figure 9: Results on unseen LLM evaluation task domains over 10 iterations (higher is better) for `Qwen2.5-7b-Instruct`. The subcaption indicates the evaluation domain. **Underlined evaluation tasks are more difficult** because the evaluation task domains are removed from the training data (i.e., OOD).

## C.4 MIXING RATIO FOUND BY DUET

We also present the mixing ratio found by DUET for our experiments after 10 BO iterations. The column title denotes the evaluation task. When the unseen task is in-domain (**TruthfulQA**), DUET automatically finds that **TruthfulQA** is relevant training domain and places more weights on it. For OOD cases, DUET automatically finds relevant training domains as well. For example, even though we do not have **gsm8k** data for training, DUET automatically finds data from **wikitext** and **sciq** more relevant in improving the performance of the trained LLM.

Table 3: Mixing ratio found by DUET in our `Llama-3-8b-Instruct` experiments after 10 BO iterations. The column indicates the unseen task domain, in the same setting and order as those found in our main experiments (Fig. 4). **NA** indicates the respective domain was not included in the training data.

| Domains | TruthfulQA | gsm8k | PubMedQA, HeadQA | CommonsenseQA, TriviaQA |
|---|---|---|---|---|
| Commonsense | 3 | 7 | 11 | **NA** |
| gsm8k | 0 | **NA** | 0 | 10 |
| headqa | 0 | 0 | **NA** | 0 |
| hellaswag | 0 | 1 | 0 | 28 |
| pubmedqa | 0 | 1 | **NA** | 0 |
| sciq | 2 | 38 | 34 | 22 |
| triviaqa | 3 | 12 | 19 | **NA** |
| wikitext | 8 | 41 | 30 | 22 |
| TruthfulQA | 84 | 1 | 6 | 18 |

# D EXTENSION AND ADDITIONAL RESULTS ON FULL-PARAMETER FINETUNING OF LLMS

While it is too computationally expensive to perform large-scale pre-training *from scratch* for now, we have performed additional experiments on **continual, full-parameter training** on 8B and 14B models (in contrast to LoRA fine-tuning used in our paper). We performed these experiments on the same setting (but we had to increase the training epoch to 3 for full-parameter training to converge better) as in Fig.4(a) and 4(b) and show the best LLM performance achieved in 10 BO iterations.

| Model Method | DUET-IF | DUET w/o IF | LESS |
|---|---|---|---|
| Llama-3-8B-Instruct | **64.3 ± 0.8** | 60.1 ± 1.1 | 51.7 ± 0.7 |
| Qwen3-14B | **73.5 ± 0.6** | 70.1 ± 0.9 | 63.2 ± 1.0 |

Table 4: Performance gain on TruthfulQA.

| Model Method | DUET-IF | DUET w/o IF | LESS |
|---|---|---|---|
| Llama-3-8B-Instruct | **85.1 ± 0.4** | 81.0 ± 0.7 | 74.4 ± 0.4 |
| Qwen3-14B | **88.6 ± 0.6** | 84.1 ± 0.9 | 80.2 ± 0.8 |

Table 5: Performance gain on gsm8k (OOD).

The results are generally consistent with our paper's finding on LLM LoRA fine-tuning, and DUET with IF performs better than its baselines. We will include this into the revised manuscript. We hope these additional results suggest that our approach is equally feasible for full-parameter fine-tuning.

One noteworthy point is that while computing IF scores for larger models is more expensive (i.e., computational cost scales with number of model parameters [1]), a practical approach is to use a smaller surrogate model to compute the IF scores, which reduces the computational time. We used the same set of IF-scores computed from the LoRA parameters here. If computational budget is a concern, we can also free to use less expensive data selection methods in DUET's inner loop (as elaborated in Sec. 3.2 of our paper) with some performance-cost tradeoff.

## E    DISCUSSION ON COMPUTATION AND MEMORY OVERHEAD OF DUET

Here, we provide a discussion of DUET-IF's computation and memory overhead.

## F    COMPUTATION OVERHEAD OF BO IN DUET

Let $T$ denote the number of Bayesian Optimization (BO) iterations and $n$ denote the dimension of the data mixture (i.e., the number of training domains). In practice, the dominant cost of DUET comes from fine-tuning the LLM $T$ times, since BO requires multiple function evaluations (Frazier, 2018). This allows BO to exploit feedback from the unseen evaluation task and avoid brute-force enumeration of all possible mixture ratios.

Beyond LLM fine-tuning, BO incurs additional computational overhead. When using a Gaussian Process (GP) surrogate, the primary cost arises from inverting the $T \times T$ kernel matrix when re-estimating GP hyperparameters using maximum likelihood (see Eq. (5.8) in (Williams & Rasmussen, 2006)). This is typically performed using a Cholesky decomposition, which costs $\mathcal{O}(T^3)$.

Next, optimizing the acquisition function at each iteration typically requires gradient-based optimization. For the UCB acquisition function used in our work, computing gradients of the GP mean and standard deviation incurs $\mathcal{O}((nT + nT^2)c)$ operations, where $\mathcal{O}(nT)$ comes from differentiating $\mu(x_{\text{candidate}})$, $\mathcal{O}(nT^2)$ from differentiating $\sigma(x_{\text{candidate}})$, and $c$ is the number of restarts used in acquisition optimization (see `botorch` acquisition optimization documentation).

**(A)** Putting these together, the total BO compute at iteration $T$ is:

$$\mathcal{O}(T^3) + \mathcal{O}((nT + nT^2)c) = \mathcal{O}(T^3),$$

since typically $n \ll t$. Summing over from the first iteration yields the same $\mathcal{O}(T^3)$ complexity because Cholesky updates allow reuse of factorizations, avoiding full recomputation at each iteration.

## G    COMPUTATION OVERHEAD OF IF SCORES IN DUET-IF

When DUET is combined with data selection methods such as Influence Functions (IF), additional compute costs arise.

**Influence Scores.**    Given $N$ datapoints and $p$ trainable model parameters (in other paper, $p$ is the number of parameters in the model LoRA), direct computation of influence scores requires

$\mathcal{O}(Np^2 + p^3)$ operations (Koh & Liang, 2017). However, stochastic estimation techniques (Sec. 3 in (Koh & Liang, 2017)) reduce this to $\mathcal{O}(Np)$. Importantly, IF scores only need to be computed once and can be reused across BO iterations.

**Other Data Selection Methods.** Different data selection strategies incur different costs, but these are typically amortized since they are computed once. For example, LESS incurs $\mathcal{O}(Nbp)$ operations, where $b$ is the number of checkpoints used (Xia et al., 2024).

Putting **(A)** and **(B)** together, the joint compute cost of DUET-IF is:

$$\mathcal{O}(T^3 + Np).$$

## H  MEMORY OVERHEAD

BO alone requires $\mathcal{O}(T^2)$ memory to store the $T \times T$ kernel matrix. During IF computation, we require $\mathcal{O}(p)$ memory to store parameter gradients; the Hessian $H$ need not be stored explicitly, as Hessian–vector products can be computed efficiently using conjugate gradient or stochastic methods (Koh & Liang, 2017). After computation, storing the IF scores requires $\mathcal{O}(N)$ memory.

Thus, DUET-IF has a total memory overhead of:

$$\mathcal{O}(T^2 + p).$$

Additionally, because DUET fine-tunes an LLM using different data mixtures across BO iterations, we maintain a copy of the best-performing LoRA adaptor throughout the optimization.

In summary, the computation overhead of DUET-IF is:

$$\mathcal{O}(T^3) + \mathcal{O}(Np),$$

where $\mathcal{O}(Np)$ can be precomputed before optimization and the memory overhead is:

$$\mathcal{O}(T^2 + p).$$

## I  MORE DETAILS ON MIXING RATIO FOUND

| ↓ Domains → Iterations | 1 | 2 | 3 | 4 | 5 | 6 | 7 | 8 | 9 | 10 | DoReMi |
|---|---|---|---|---|---|---|---|---|---|---|---|
| commonsenseQA | 11 | 0 | 0 | 11 | 28 | 0 | 11 | 80 | **3** | 0 | 14 |
| gsm8k | 11 | 0 | 0 | 9 | 0 | 0 | 10 | 16 | **0** | 90 | 4 |
| headQA | 11 | 0 | 0 | 0 | 0 | 0 | 7 | 0 | **0** | 0 | 6 |
| hellaswag | 11 | 0 | 0 | 13 | 0 | 2 | 0 | 0 | **0** | 0 | 3 |
| pubmedqa | 11 | 0 | 0 | 8 | 0 | 0 | 6 | 4 | **0** | 0 | 9 |
| sciq | 11 | 0 | 0 | 0 | 0 | 0 | 16 | 0 | **2** | 10 | 14 |
| triviaQA | 11 | 0 | 0 | 0 | 60 | 17 | 0 | 0 | **3** | 0 | 20 |
| wikitext | 11 | 0 | 38 | 11 | 0 | 10 | 0 | 0 | **8** | 0 | 22 |
| truthfulQA | 11 | 100 | 62 | 48 | 12 | 71 | 50 | 0 | **84** | 0 | 18 |
| **LLM Performance** | 47 | 52 | 53 | 54 | 48 | 57 | 51 | 58 | **60(*)** | 40 | 51 |

In the table below (for clarity reasons, the numbers are rounded), we show the data mixing ratio found by DUET-IF at each iteration as compared to that found by DoReMi for the in-domain TruthfulQA task in **Fig. 4(a)** (to bridge the discussion point we raised above). We also show the data mixing ratio found by DoReMi.

We used a uniform data mixture ($\sim 11\%$ of data to each data domain) as the initial data mixture for BO. By chance (since BO with a confidence-based acquisition function tends to explore boundary inputs initially), it finds that placing more emphasis on the TruthfulQA data domain yields better LLM performance. After some adjustments, in the 9th iteration, the best performing data mixture was found. In fact, in some of our exploratory process, we found that if we increased the number of iterations beyond 10, we can find even better training data mixtures, but the gain in performance typically plateaus (as in many BO applications).

**Comparison to DoReMi:** We can see that DoReMi adopts a distributionally robust approach and allocates mixture weights more uniformly across different domains (since it cannot exploit the task feedback to infer that truthfulQA data is more relevant). This is clearly suboptimal because it is not optimized specifically towards the evaluation task. and hence its data mixture does not perform as well as DUET-IF, as shown in our experiments.

