# OpenReview forum: "DUET: Optimizing LLM Training Data Mixtures via Noisy Feedback from Unseen, Downstream Evaluation Tasks"
_ICLR.cc/2026/Conference — ICLR 2026 Poster_

### Official Review · Reviewer_QVFn · 2025-10-26

**Soundness:** 4
**Presentation:** 3
**Contribution:** 3
**Rating:** 8
**Confidence:** 4

**Summary:**

Optimizing the deployment of the model without an instance-wise loss function but only at a model-wise loss function is an important problem. The paper formalizes the setting, focuses on optimizing only data selection, and shows that this problem can be divided into two steps: optimizing a data mixture ratio and optimizing a dataset meeting this ratio. The author introduces DUET to use BO for the first optimization problem and IF-driven estimators for the second problem. Cumulative regret results for DUET is provided. DUET performs better than other baselines on LLM benchmarks with small LMs (<=8B).

**Strengths:**

- The problem formulation is clear and the writing is good.
- The introduction of the two components of DUET to solve the problem is very natural.
- Theoretical results on cumulative regrets are provided.

**Weaknesses:**

- The theoretical results don't seem very novel and more like applying previous results in the literature in a new setting.

**Questions:**

- Can we extend the set of optimization variables to be more than just the data mixing ratio (e.g., optimizing a small set of parameters of the LLM) or is the setting too noisy to reliably do so? I don't see an explicit dependence on $n$, the dimension of $r$ in Theorem 4.1.
- Can we get Theorem 4.1 without an IF-driven estimator? If so, having that somewhere would help showing how IF helps DUET theoretically?
- Does the unseen evaluation task setting mean that you train on all datasets except for that task? Also, is it true the evaluation is just one feedback (good/bad model) at the model level without access to the exact evaluation loss? These details should be clearer. You should not point to LLM-harness and be done with it.

---

> ### Author Response · Authors · 2025-11-24
> **Response Part 1/2**
>
> Thank you for the review and questions. We appreciate that you find our problem formulation "clear" and the combination of DUET's two components "very natural" in solving the problem. We hope our responses clarify the questions raised.
>
> > **Can we extend the set of optimization variables to be more than just the data mixing ratio (e.g., optimizing a small set of parameters of the LLM) or is the setting too noisy to reliably do so?**
>
> Good question. We definitely can extend the set of optimization variables to be more than the data mixing ratio. For example, we can include training hyperparameters (e.g., learning rate) and optimize them jointly together with the data mixing ratio. The noisy setting is not an issue because BO can handle them gracefully both theoretically and empirically (in our experiments, LLM fine-tuning and inference is a noisy process, but DUET-IF still works well as compared to other baselines). In our paper, we choose to focus on data-related variables (i.e., data mixing ratio) because we want to investigate how we can interleave different data selection methods into BO elegantly. Your suggestion is certainly a good future research direction.
>
> > **Can we get Theorem 4.1 without an IF-driven estimator? If so, having that somewhere would help showing how IF helps DUET theoretically?**
>
> Thank you for the suggestion. Theorem 4.1 states the regret bound of DUET with the IF-driven estimator; this estimator incurs an observation noise given in Theorem 3.2.
>
> The neatest way to rephrase Theorem 4.1 with and without an IF-driven estimator (or any other data selection alternatives that incur different kinds of estimation noise) is to quantify the difference in observation noise in the BO process. This is because in our setting the estimation error different estimators directly changes the observation noise in BO.
>
> For instance, if we used random sampling without an IF-driven estimator, our estimation error becomes larger and more tail-end (See Fig. 3 for empirical comparison). This implies our observation noise distribution is governed by larger parameters $c$ and $\lambda$. A larger $c$ then leads to larger $A_{c,k}$ (in Theorem 4.1), causing our BO regret bounds to be larger. Exactly how much $c$ and $\lambda$ is larger depends on the problem setting. One way we can quantify this approximately is to empirically observe the estimation error distribution difference with and without an IF-driven estimator in Fig. 3.
>
> We will include these discussions into our revised manuscript and make it clearer that with and without the IF-driven estimator (or any other data selection method), the term $A_{c,k}$ scales w.r.t. different selection methods and influences our regret bound. We hope this has answered your question.
>
> > **The theoretical results don't seem very novel and more like applying previous results in the literature in a new setting.**
>
> While our overall backbone of theoretical regret bound derivation follows from existing literature, **drawing the direct connection between how different data selection methods in DUET's inner loop influence Bayesian optimization's observation noise, and consequentially its regret, is novel.**
>
> Specifically, our work is the first to empirically observe the noisy process behind data selection methods (Fig. 3) in DUET's inner loop and capture the error of this process with a truncated exponential distribution (Theorem 3.2), before studying how this error distribution influences BO's convergence in Theorem 4.1.

---

> ### Author Response · Authors · 2025-11-24
> **Response Part 2/2**
>
> > **Does the unseen evaluation task setting mean that you train on all datasets except for that task?**
>
> Thank you for the question. Let us clarify: to be precise, the unseen evaluation task setting indicates that we do not have any fine-grained data information surrounding the evaluation task (or to identify what evaluation task it is). Hence, we can only utilize coarse, performance feedback to iteratively improve the training data mixture. It is still possible that the evaluation task data is part of the training data domain (but we do not know this directly), and by exploiting feedback, we can infer whether to place emphasis on that data domain in our data mixture.
>
> In Fig. 4(a), we allowed the evaluation task data domain to be part of the training data domains. In Fig. 4(b), (c) & (d), we considered the out-of-domain (OOD) setting, where we _remove the unseen evaluation task data domain_ explicitly from the training data domains. In this case, DUET needs to infer the optimal data mixtures that train an LLM well for an OOD evaluation task. We considered the OOD setting in our experiments make our evaluation settings more difficult for DUET. However, as seen from our empirical results, DUET still performs well in such OOD settings.
>
> We hope our response has answered the reviewer's question. Please let us know if you need any other clarifications.
>
> > **Also, is it true the evaluation is just one feedback (good/bad model) at the model level without access to the exact evaluation loss? [...]**
>
> We appreciate the feedback and we will include the following details into our revised manuscript: After fine-tuning the LLM, we subject it to the evaluation task procedure given by `lm-evaluate-harness` (https://github.com/EleutherAI/lm-evaluation-harness). This library subjects the fine-tuned LLM to an evaluation test question bank (e.g., gsm8k, TriviaQA) containing a fixed number of questions and evaluate the percentage of questions in which the LLM answered correctly. In some settings (Fig.4(c) and (d)), we combined two domains and took the average task accuracy as the evaluation task performance. Throughout the evaluation, we assume no access/knowledge of the evaluation token-wise loss or evaluation data. We hope this helps in addressing your feedback. If you find the response satisfactory, we will include it in the final version of our manuscript.
>
> ___
>
> Once again, thank you for reviewing our paper and we hope our responses have improved your opinion of our work. Please let us know if you need further clarifications, and if our responses are satisfactory, we will include them in the final version of our manuscript.

---

> > ### Comment · Reviewer_QVFn · 2025-11-24
> >
> > Thank you the authors for the clarification! I have no questions for now.

---

### Official Review · Reviewer_6k1g · 2025-10-29

**Soundness:** 2
**Presentation:** 3
**Contribution:** 2
**Rating:** 4
**Confidence:** 2

**Summary:**

This paper presents DUET, a method that combines data mixture optimization and data selection, for LLM fine tuning. The authors propose a two stage optimization, in which the inner problem takes a mixture ratio as input and samples data satisfying the mixture ratio via data selection, and the outer problem solves the mixture ratio taking the optimization of the inner problem as a blackbox. In particular, the outer problem is solved by Bayesian optimization.

The authors conduct extensive experiments to demonstrate the effectiveness of the proposed method.

**Strengths:**

1. The paper is generally well-written, clear, and easy to follow. The proposed method is elegant and easy to implement.

2. The empirical results strongly support the effectiveness of the proposed method.

**Weaknesses:**

1. The proposed method is a simple combination of data selection and Bayesian optimization.

2. The reviewers have some questions about the empirical results.

**Questions:**

1. The authors highlight that the experiments are done with sampling size of k = 1 in each BO iteration. However, the results seem too good to be true, given that the optimization is done over a degree-8 simplex. Note that almost all experiments in Figure 4 show >10-point improvements within 10 total samples. Moreover, in 4(a), there are >10-point improvements even with only 1 sample, and if the reviewer understands correctly how Bayesian optimization works, with only 1 sample, Bayesian optimization cannot provide useful suggestions. The reviewer may be missing something here. Could the authors help clarify?

2. Related to 1 above, could the authors provide the mixture found by the proposed methods for each iteration?

3. It seems that this paper mainly applies Bayesian optimization to the data mixture optimization problem (along with data selection), which limits the technical contributions from this paper. Are there technical difficulties the authors could highlight? In particular, are there modifications needed for Bayesian optimization in order to address challenges from "noisy feedback from unseen evaluation tasks"?

---

> ### Author Response · Authors · 2025-11-24
> **Response Part 1/2**
>
> Thank you for reviewing our paper. We are glad that you find our method "elegant" and that our "empirical results strongly support the effectiveness of the proposed method." We have provided our responses to your comments below and we hope they improve your opinion of our work.
>
> >**if the reviewer understands correctly how Bayesian optimization works, with only 1 sample, Bayesian optimization cannot provide useful suggestions. The reviewer may be missing something here. Could the authors help clarify?**
>
> Good question. While BO is unlikely to have good results at the initial samples/iterations, in Fig. 4(a) we are looking at the TruthfulQA task with in-domain training data (meaning TruthfulQA data domain is available in the training domains). By chance, in the second iteration, the acquisition function proposed a mixing ratio that placed heavy emphasis on the TruthfulQA data domain, and therefore unsurprisingly the fine-tuned LLM performed well on the TruthfulQA task (since our training data contained mostly TruthfulQA data). If we looked at the other tasks in Fig. 4(b), (c) and (d), the LLM performance remained relatively stagnant (due to BO exploration) at the beginning before better data mixtures are found.
>
> We hope this clarifies the reviewer's question. Please let me know if you need further clarifications. In addition, we will also provide and discuss the data mixtures found at every iteration in the later section.
>
> > **[...] 10-point improvements within 10 total samples**
>
> In our original paper, we have provided an example of a real optimization landscape in a lower-dimensional problem (Fig. 2). We can see that the high-performing region is concentrated around one region and hence, the landscape is not overly-complicated (unlike complex toy-functions like the Ackley function) and we speculate this allows DUET to perform well in 10 iterations. We are not implying that 10 BO iterations can produce the _optimal_ data mixture; rather, we intend to show that 10 iterations is _sufficient_ for DUET to produce better data mixtures than other baselines.
>
> > **Related to the point above, could the authors provide the mixture found by the proposed methods for each iteration?**
>
> Thank you for the comment.
>
> In the table below (for clarity reasons, the numbers are rounded), we show the data mixing ratio found by DUET-IF at each iteration as compared to that found by DoReMi for the in-domain TruthfulQA task in **Fig.4(a)** (to bridge the discussion point we raised above). We also show the data mixing ratio found by DoReMi, a baselines data-mixing method. Initially, we used a uniform data mixture (~11% of data to each data domain) in DUET-IF. By chance (since BO with a confidence-based acquisition function tends to explore boundary areas initially), it finds that placing more emphasis on the TruthfulQA data domain yields better LLM performance. After some adjustments, in the 9th iteration, the best performing data mixture was found. In fact, in some of our exploratory process, we found that if we increased the number of iterations beyond 10, we can find even better training data mixtures, but the gain in performance typically plateaus (like in many BO applications).
>
> | Data domain  &#8595; Iterations of DUET &#8594; | 1  | 2   | 3  | 4  | 5  | 6  | 7  | 8  | **9(*)**  | 10 | Mixture found by DoReMi, another baseline|
> | ------------- | -- | --- | -- | -- | -- | -- | -- | ------ | ------ | -- | -|
> | commonsenseQA | 11 | 0   | 0  | 11 | 28 | 0 | 11 | 80 | **3**  | 0  |14|
> | gsm8k         | 11 | 0   | 0  | 9 | 0  | 0  | 10 | 16 | **0**  | 90 |4|
> | headQA        | 11 | 0   | 0  | 0 | 0  | 0 | 7  | 0  | **0**  | 0  |6|
> | hellaswag     | 11 | 0   | 0 | 13 | 0  | 2 | 0  | 0  | **0**  | 0  |3|
> | pubmedqa      | 11 | 0   | 0  | 8 | 0  | 0  | 6  | 4  | **0**  | 0  |9|
> | sciq          | 11 | 0   | 0  | 0  | 0  | 0  | 16 | 0  | **2**  | 10 |14|
> | triviaQA      | 11 | 0   | 0  | 0  | 60 | 17 | 0  | 0  | **3**  | 0  |20|
> | wikitext      | 11 | 0   | 38  | 11  | 0  | 10 | 0  | 0  | **8**  | 0  |22|
> | truthfulQA    | 11 | 100 | 62  | 48 | 12 | 71 | 50 | 0  | **84** |  0  |18|
> | LLM Performance   |  47  |  52   |  53   | 54   |  48  |  57  |  51  |  58  |   **60**     |  40   |51|
>
>
> **Comparison to DoReMi:** Doremi adopts a distributionally robust approach and allocates mixture weights more uniformly across different domains (since it cannot exploit the task feedback to infer that truthfulQA data is more relevant). This is clearly suboptimal and we see that the data mixture fund does not perform as well as DUET-IF, as seen from the fine-tuned LLM performance indicated in the last row.
>
> Thanks again for raising this point, we hope this response has provided some clarity. We have included the additional mixing ratios and this discussion in Appendix D (under Rebuttal additions) of our revised manuscript. If the reviewer would like to visualize the BO search process in a smaller-dimensional simplex, please refer to Fig. 2 of our paper.

---

> ### Author Response · Authors · 2025-11-24
> **Response Part 2/2**
>
> > **Are there technical difficulties the authors could highlight? In particular, are there modifications needed for Bayesian optimization in order to address challenges from "noisy feedback from unseen evaluation tasks"? [...] The proposed method is a simple combination of data selection and Bayesian optimization.**
>
> Thank you for the question. There are several novel aspects of DUET that distinguish it from standard Bayesian Optimization (BO) or data selection. In our setting, the interaction between BO and data selection introduces technical challenges that our paper addresses - specifically, how data selection theoretically and empirically helps to improves the effectiveness of BO in data mixing. We explain these challenges and our contributions below.
>
> First, even if we directly apply BO to optimize the data mixing ratio
> $r$, it is not obvious how to construct a good training mixture that matches the queried mixing ratio at each BO iteration. A naive approach that randomly samples data points from each domain to satisfy the proposed ratio turns out to be extremely noisy. As we show in Sec. 3.2, the fine-tuned LLM performance obtained from such randomly constructed mixtures exhibits a large and heavy-tailed estimation error distribution. We found that this noise enters the BO algorithm as large observation noise and makes it significantly harder for BO to identify a good training data mixture (i.e., to solve Eq. (2)).
>
> Second, DUET’s key insight is that the choice of data selection method directly controls the observation noise in BO. This is not merely a heuristic but also can be quantified rigorously: In Theorem 3.2, we formally quantify how data selection can reduce the estimation error distribution to a truncated exponential distribution. The choice of a truncated exponential distribution comes from our empirical observation (Fig. 3) that data selection reduces both the variance and tail-heaviness of the estimation error in this way. In Theorem 4.1, we then analyze how this reduced noise leads to provably faster BO convergence (i.e., reducing the cumulative regret, allowing BO to converge quickly). We also verified the effectiveness of different data selection methods (varying from Influence scores, LESS, Diversity) interleaved into DUET in our empirical results, and found DUET to outperform existing baselines (e.g., plain data selection or mixing).
>
> In summary, DUET addresses a key technical challenge of noisy data mixture construction and develops a principled approach that both reduces this noise and improves BO convergence. This is supported by theoretical guarantees and empirical evidence in our paper. Our paper also found that different data selection methods have different cost-benefit tradeoffs, and our algorithm, DUET, is flexible enough to incorporate any of them into the BO procedure - this flexibility is a major strength of our paper and makes DUET simple and elegant to use for real-world practitioners.
>
> We hope this explanation clarifies and summarizes the technical contributions and insights behind DUET.
>
> ___
>
> Once again, thank you for reviewing our paper and we hope our responses have improved your opinion of our work. Please let us know if you need further clarifications, and if you find our responses satisfactory, we will include them into the final version of our manuscript.

---

### Official Review · Reviewer_tBBS · 2025-11-01

**Soundness:** 3
**Presentation:** 2
**Contribution:** 3
**Rating:** 6
**Confidence:** 4

**Summary:**

This paper proposes DUET, a novel global-to-local data mixture optimization algorithm for LLM finetuning, especially when the target evaluation task is unseen and with noisy performance feedback. Theoretically, the authors applies Bayesian Optimization (BO) for global (inter-domain) mixture rate search, combined with a local (inner-domain) sample-level search by influence function based methods. Empirically, the proposed method demonstrates an improvements on the performance on in-domain and out-of-domain tasks, compared to several existing baselines.

**Strengths:**

1. The paper targets at a critical and challenging problem of data mixture optimization, under a practical and real-world scenario, when the target task is unseen and with noisy feedbacks.
2. The paper proposes a bayesian optimization based method for data mixture search, which serves as a reliable estimation with coarse feedbacks from the target, black-box-like tasks.
3. The proposed DUET method obtains a significant performance gain, outperforming several existing baselines.

**Weaknesses:**

1. The computational cost and overhead have not been discussed. Both the retraining involved in the BO process and the influence function computation can be computational intensive, especially when the model gets scaled-up (e.g. 70B+). can you provide an analysis on computation and memory overhead caused by the BO and BO+IF methods?

2. To compute the influence scores (IF), do you require the validation set from the target distribution? if so, how do you ensure the IF scores are reliable under the unseen and noisy evaluation environment? Also, the data may not be available if the evaluation system is a black-box.

3. The effectiveness of BO depends on how well the Gaussian Process models the objective function.

**Questions:**

1. can you provide an analysis on computation and memory overhead caused by the BO and BO+IF methods?

2. How does the BO method scales by the number of data domain $k$?

3. To compute the influence scores (IF), do you require the validation set from the target distribution? if so, how do you ensure the IF scores are reliable under the unseen and noisy evaluation environment? Also, the data may not be available if the evaluation system is a black-box.

---

> ### Author Response · Authors · 2025-11-24
> **Response Part 1/3**
>
> We appreciate your comment that our work "targets at a critical and challenging problem of data mixture optimization, under a practical and real-world scenario". We have provided responses to your questions and we hope they improve your opinion of our work.
>
> > **[...] Can you provide an analysis on computation and memory overhead caused by the BO and BO+IF methods?.**
>
> Thank you for the feedback. We have provided an analysis of the computation and memory overhead of DUET (containing BO + a data selection method in its inner loop) here. We have also included them in App. D.2 of our revised manuscript.
>
> **Computation overhead of BO in DUET**
>
> Let $T$ be the number of BO iterations and $n$ be the number of training domains in our training data mixture. Practically, fine-tuning the LLM $T$ times is the most time-consuming part of DUET due to the BO's iterative nature, but collecting multiple evaluations is what's necessary in BO [5] and furthermore, re-finetuning or re-training an LLM over extended period of time aligns with the deployment life cycle of many LLMs such as ChatGPT, Gemini, DeepSeek, which are all iteratively fine-tuned over time (the optimization is done over a long period, instead of all at once). In addition, BO is sample-efficient and reduces the need to evaluate all possible mixing ratios in a brute-force manner, and is the only way that allows us to exploit feedback from the unseen evaluation task.
>
> We will now discuss the additional computation overhead (apart from fine-tuning the LLM) associated with BO. The key computation cost of BO that uses a GP as the surrogate model arises from the need to invert the $T \times T$ kernel matrix when we use maximum-likelihood to derive the GP hyperparameters based on $T$ historical observations (See Eq. (5.8) from [1]) at every iteration. This is typically done with Cholesky decomposition.
>
> Next, optimizing the acquisition function at each iteration is typically done with gradient descent of the mean and std. of the GP for the UCB acquisition function used in our work. This operation's computation cost incurs $\mathcal{O}((nT + nT^2)c)$ computation cost, where $\mathcal{O}(nT)$ comes from computing the gradient of $\mu(x_{\text{candidate}})$ and $\mathcal{O}(nT^2)$ comes from computing the gradient of $\sigma(x_{\text{candidate}})$ in the acquisition function indicated in line 3 of Algorithm 1 in our paper. The constant term $c$ indicates the pre-decided number of optimization restarts commonly used in gradient optimization (see https://botorch.org/docs/optimization).
>
> Put together, if we simply used BO with no data selection methods in the inner loop, we incur a computation cost of $\mathcal{O}(T^3) + \mathcal{O}(nT + nT^2)c \approx \mathcal{O}(T^3)$ at the T-th iteration (since $n$ is typically smaller than $T$). Even if we sum this operation from the first iteration, the cost is still $\mathcal{O}(T^3)$ because we can apply Cholesky updates to the decomposition from the previous iteration to avoid re-inverting the $T \times T$ kernel matrix from scratch every iteration.
>
> **Computation overhead of IF scores in DUET-IF**
>
> Next, different data selection methods have different compute cost when interleaved into DUET.
>
> **Influence scores**: assuming we have $N$ data points and $p$ model parameters, pre-computing the influence scores directly requires $\mathcal{O}(Np^2+p^3)$ operations [2]. However, this operation can be sped up (Sec. 3. in [2]) to $\mathcal{O}(Np)$ via stochastic estimation [2]. Notably, Influence scores can be _pre-computed_ and needs to be only computed once. Hence, **the total computation cost of BO + IF is $\mathcal{O}(t^3 + Np)$.**
>
> **Other data selection methods**: Different data selection methods have different compute overhead. Most of these compute overhead is amortized such that we only need to compute these once. For instance, LESS requires a compute overhead of $\mathcal{O}(Nbp)$ (where $b$ is the number of checkpoints used [3]).
>
> _[Continued in next response 2/3]_

---

> ### Author Response · Authors · 2025-11-24
> **Response Part 2/3**
>
> _[Continued from previous part 1/3]_
>
> **Memory overhead**
>
> BO alone uses $\mathcal{O}(T^2)$ memory to store the $T \times T$ kernel matrix of past observations. On the other hand, during IF score computation, we require additional $\mathcal{O}(p)$ memory because we need to store $p$ parameter gradients (in this case $p$ is the number of parameters in the LoRA adaptors). We do not need to store the $p\times p$ Hessian matrix $H$ directly because we can use Hessian–vector products in approximation methods such as conjugate gradients or stochastic estimation [2]. After the IF scores are computed, they are stored using $\mathcal{O}(N)$ memory. Therefore, using BO + IF requires $\mathcal{O}(T^2 + p)$ memory overhead. In addition, since we are fine-tuning an LLM with different data mixtures iteratively, we need to store a copy of the best-performing LoRA adapter at all times, so that we can use them at the end of the optimization process.
>
> To summarize, the computation overhead of DUET-IF is $\mathcal{O}(T^3) + \mathcal{O}(Np)$ plus the need to fine-tune the LLM $T$ times. The memory overhead of DUET-IF is $\mathcal{O}(T^2 + p)$. We have included this discussion in App. D.2 in the revised manuscript. We hope this response has helped clarify your question on the computation and memory overhead of DUET-IF.
>
> > **How does the BO method scales by the number of data domain?**
>
> The number of data domains corresponds to the BO problem dimension directly. In general, the effectiveness of BO diminishes when the problem dimension becomes very large without any special assumptions. However, in our case, where the number of data domains is 8, our experiments showed that we were able to achieve good LLM performance in a reasonable number of BO iterations (fewer than 10).
>
> There are two reasons why BO becomes less effective with higher problem dimension. _First_, it becomes more difficult to use gradient descent to globally optimize the acquisition function at every BO iteration. _Second_, theoretical results show that the cumulative regret garnered by BO at each BO iteration $T$ increases with problem dimension $d$ by a factor of $\gamma_T = \mathcal{O}((ln T)^{d+1})$ [5] for the SE kernel. Specifically, this factor refers to the maximum information gain (indicating the complexity of the problem) and appears in our cumulative regret proof under Eq. (13). If $d$ is larger, BO's cumulative regret will increase. We hope this answers the reviewer's question.
>
> > **To compute the influence scores (IF), do you require the validation set from the target distribution? if so, how do you ensure the IF scores are reliable under the unseen and noisy evaluation environment? Also, the data may not be available if the evaluation system is a black-box.**
>
> This is a great question. As mentioned in Section C.1, Line 1182 onward, we calculated the IF scores by isolating a subset of data as the validation set from the training data (not the evaluation task). This means, our IF scores are computed _within_ the training data domains, instead of w.r.t. the evaluation task distribution. So, we do not need the data from the "black-box evaluation system", as you have pointed out, since our setting is such that we do not have access to any data information regarding our evaluation task. Despite so, the reason why these scores are still "reliable under the unseen environment" is because low IF-score data are typically nonsensical/low-quality text that are unlikely to be useful for _any_ evaluation tasks. Hence, even though the scores are computed from a different distribution, we can still use them to down-sample/filter out nonsensical data, obtaining a higher-quality training data mixture and yielding good performance in DUET.
>
> We hope this addresses the reviewer's questions. Please let us know if you need further clarifications.

---

> ### Author Response · Authors · 2025-11-24
> **Response Part 3/3**
>
> > **The effectiveness of BO depends on how well the Gaussian Process models the objective function.**
>
> Thank you for the comment. Fig. 2 in our original paper empirically plots the objective function landscape in a smaller-dimensional problem (We had to brute-force evaluate every single data mixing ratio for illustration purposes). The objective function empirically indicates that it is approximately smooth/continuous, which GP can model well. In fact, prior works [6] have also demonstrated that Bayesian Optimization still works well even if the objective function is not perfectly smooth. We hope this addresses the reviewer's comment.
>
> ___
>
> Once again, thank you for reviewing our paper and we hope our responses have improved your opinion of our work. Please let us know if you need further clarifications, and if you find our responses satisfactory, we will include them into the final version of our manuscript.
>
> [1] C. E. Rasmussen & C. K. I. Williams, Gaussian Processes for Machine Learning, 2006
>
> [2] Pang Wei Koh, Percy Liang, Understanding Black-box Predictions via Influence Functions, 2017.
>
> [3] Mengzhou Xia, Sadhika Malladi, Suchin Gururangan, Sanjeev Arora, Danqi Chen, LESS: Selecting Influential Data for Targeted Instruction Tuning, 2024.
>
> [4] Kandasamy, K., Schneider, J., and Póczos, B.High dimensional Bayesian optimisation and bandits via additive models. 2015.
>
> [5] Sayak Ray Chowdhury, Aditya Gopalan, On Kernelized Multi-armed Bandits. 2017
>
> [6] Colliandre et al. Bayesian Optimization in Drug Discovery. 2024.

---

### Official Review · Reviewer_9jKt · 2025-11-10

**Soundness:** 3
**Presentation:** 2
**Contribution:** 2
**Rating:** 6
**Confidence:** 4

**Summary:**

The paper proposes DUET, a data selection method for optimizing the data mixture used in language model fine-tuning. DUET leverages feedback from downstream evaluation tasks to guide its selection process. The method combines Bayesian optimization with influence function to identify the optimal data composition. The authors validate DUET's effectiveness through comprehensive empirical evidence and theoretical analysis.

**Strengths:**

1. Optimizing the data mixture with "evaluation aware"  for language model fine-tuning is a practical and important field, primarily because the ideal data distribution is rarely known and language models are intrinsically multitask learners.
2. The authors propose a novel method to find an optimal data mixture, using Bayesian optimization for the outer optimization loop and influence functions for the inner optimization.
3. The authors support their approach with sound theoretical analysis and comprehensive experimental results.

**Weaknesses:**

1. The method is computationally intensive, requiring multiple separate LLM fine-tuning runs to gather feedback for the Bayesian Optimization loop.
2. The reliance on IF is questionable, as some of works point out IF for LLM is not very reliable [1].
2. The experiments are confined to fine-tuning. The approach's feasibility for large-scale pre-training, especially for computationally heavy estimators like IF, is not demonstrated.

[1] Basu, Samyadeep, Philip Pope, and Soheil Feizi. "Influence functions in deep learning are fragile." arXiv preprint arXiv:2006.14651 (2020).

**Questions:**

Please see weaknesses.

---

> ### Author Response · Authors · 2025-11-24
> **Response Part 1/2**
>
> Thank you for reviewing our work. We appreciate your recognition of our paper's practical evaluation setting and "sound theoretical analysis and comprehensive experimental results". We hope that our responses will clarify any remaining doubts, and thank you in advance for engaging us.
>
> ___
>
> > **The method is computationally intensive, requiring multiple separate LLM fine-tuning runs to gather feedback for the Bayesian Optimization loop.**
>
> Thank you for the feedback. We would like to emphasize that in our problem setting, only coarse and noisy feedback is available from the downstream evaluation task to improve the training data mixture. Hence, in our setting, arguably the only way to optimize the training data mixture for the task is to observe the task feedback and fine-tune the LLM repeatedly across multiple feedback iterations. Existing data selection baselines are not suitable because they typically require fine-grained data information related to the evaluation task, which is unavailable here (and in our experiments, we showed that they do not perform well). This setting is also reflective of the real-world, since subjecting LLMs to multiple rounds of fine-tuning is a natural part of the deployment life-cycle to improve customer-facing LLMs (e.g., ChatGPT, Gemini, and Deepseek are progressively improved over time).
>
> Under this setting, the simplest, and most naive approach is to iterate through all possible training data mixtures, fine-tune the LLM, and evaluate the LLM performance on the downstream evaluation task in a brute-force manner. However, this is clearly too computationally expensive. The use of Bayesian Optimization (BO) in our work actually _improves_ the sample-efficiency of the optimization problem, reducing the number of times we need to collect feedback from the evaluation task to optimize our training data mixture (In our experiments, 10 BO iterations is sufficient for us to outperform other baselines). Therefore, our approach, DUET actually reduces the number of feedback rounds needed to find the optimal training data mixture, and should be considered efficient as compared to other relevant approaches. We view this as a strength, rather than a weakness.
>
> We hope this has clarified the reviewer's question and if the reviewer is satisfied with our response, we will add it to our revised manuscript.
>
> > **[...] some of works point out IF for LLM is not very reliable [...]**
>
> Thank you for raising this point. While IF scores can be brittle in certain scenarios (e.g., it cannot pinpoint important data precisely or its computation is brittle), its key advantage is that it works relatively well in the converse case: identifying data points which are noisy and not relevant to the evaluation set at all (e.g., nonsensical pieces of text for language tasks). This makes IF useful in our setting because even if we do not know the evaluation task, nonsensical datapoints with low-IF scores are unlikely to be useful for _any_ unseen task, and they will be down-sampled and removed from the resulting data mixtures if used in DUET. For more discussion surrounding the use of IF as compared to other data selection methods used in DUET's inner loop, feel free to check out A.5, A.6 and specifically lines 789-795 of our paper, where we have discussed different data selection methods and their performance-cost tradeoffs.
>
> Regarding possible brittle computation of IF-scores, the library we used (https://github.com/alstonlo/torch-influence) has three ways to compute IF-scores: Linear time Stochastic Second-Order Algorithm (LiSSA), conjugate gradient, and exact computation. LiSSA is the fastest and is still relatively stable as long as hyperparameters are chosen properly i.e., those recommended in [1]. Conjugate gradient computation is more stable and still relatively efficient (see C.1 for exact computation time of DUET). By following these guidelines, we found our IF-scores relatively computation stable in practice.
>
> Lastly, the choice of the data selection method and DUET are _complementary_ in nature. So, we are not restricted to using IF-scores in DUET's inner loop, and other data selection choices are also possible (each with different compute-performance tradeoff). We have outlined different computation time of different data selection methods (LESS, Diversity etc) in C.1, alongside their pros and cons in A.5. Therefore, the flexibility of DUET means that practitioners are not restricted to using IF scores and are free to choose other data selection methods.
>
> Thanks again for raising this point. We hope this response has helped clarify your question.

---

> ### Author Response · Authors · 2025-11-24
> **Response Part 2/2**
>
> > **The experiments are confined to fine-tuning. The approach's feasibility for large-scale pre-training, especially for computationally heavy estimators like IF, is not demonstrated.**
>
> While it is too computationally expensive to perform large-scale pre-training _from scratch_ for now, we have performed additional experiments on continual, **full-parameter** training on 8B and 14B models (in contrast to LoRA used in our paper). We performed these experiments in the same setting (we needed to increase the training epoch to 3 for full-parameter training to converge reasonably) as in Fig.4(a) and 4(b) and show the best LLM performance achieved in 10 BO iterations. We have included the additional results and discussions in Appendix D (under Rebuttal additions).
>
> Full-parameter fine-tuning performance gain for TruthfulQA:
>
> |Model Method| DUET-IF | DUET without IF | LESS |
> |--|-------|-------|-------|
> |Llama-3-8B-Instruct| **64.3 $\pm$ 0.8** | 60.1 $\pm$ 1.1 | 51.7 $\pm$ 0.7 |
> |Qwen3-14B| **73.5 $\pm$ 0.6** | 70.1 $\pm$ 0.9 | 63.2 $\pm$ 1.0 |
>
> Full-parameter fine-tuning performance gain for gsm8k (OOD):
>
> |Model Method| DUET-IF | DUET without IF | LESS |
> |--|-------|-------|-------|
> |Llama-3-8B-Instruct| **85.1 $\pm$ 0.4** | 81.0 $\pm$ 0.7 | 74.4 $\pm$ 0.4 |
> |Qwen3-14B| **88.6 $\pm$ 0.6** | 84.1 $\pm$ 0.9 | 80.2 $\pm$ 0.8 |
>
> The results are generally consistent with our paper's finding on LLM LoRA fine-tuning, and DUET-IF performs better than its counterparts and baselines. We have included this into App. D of the revised manuscript. We hope these additional results further show that our approach is equally feasible for full-parameter fine-tuning.
>
> One noteworthy point is that while computing IF scores for larger models is more expensive (i.e., computational cost scales with number of model parameters [1]), a practical approach is to use a smaller surrogate model to compute the IF scores, which reduces the computational time. If computational budget is a concern, we are also free to use less expensive data selection methods in DUET's inner loop (as elaborated in Sec. 3.2 of our paper) with some performance-cost tradeoff.
>
> ___
>
> Once again, thank you for reviewing our paper and we hope our responses have improved your opinion of our work. Please let us know if you need further clarifications, and if you find our responses satisfactory, we will include them into the final version of our manuscript.
>
> [1] Koh et al. Understanding Black-box Predictions via Influence Functions. 2017.
>
> [2] Agarwal et al. Second-Order Stochastic Optimization for Machine Learning in Linear Time. 2017.

---

### Author Response · Authors · 2025-12-02
**Summary of rebuttals**

Dear ACs and reviewers,

We appreciate the AC and all reviewers for taking time to read our paper. Our algorithm, DUET, addresses a unique problem setting of optimizing training data mixtures for an unseen evaluation task performance based on only noisy feedback from the task. Our work formalizes this problem as a two-stage optimization problem. By drawing connection between Bayesian Optimization (BO) with noise reduction of different data selection methods, we _empirically and theoretically_ show that using data selection methods in the optimization inner loop reduces the observation noise under the BO framework, improving convergence rates to optimal data mixtures.

**Although the discussion phase ended earlier and we did not receive responses from most reviewers**, we are encouraged to receive initial positive comments (and mostly positive scores) for our work:

1. Our work targets a "practical and important" problem setting of evaluation aware language model fine-tuning, which addresses the optimization of training data for unseen and noisy evaluation tasks (where the "ideal data distribution or fine-grained information of a target task is unknown", which is very relevant in real-world applications e.g., conversation data are end-to-end encrypted). [`9jKt`, `tBBS`, `QVFn`] No existing works have addressed this important setting.
2. Our work's clear writing and problem formulation [`QVFn`], DUET's elegant [`6k1g`] and novel [`9jKt`, `tBBS`] interleaving of Bayesian optimization with different data selection methods, alongside theoretical proof of regret bounds that show that DUET converges to the optimal data mixture. [`9jKt`, `QVFn`]
3. Our experimental results [`9jKt`, `tBBS`, `6k1g`], which showcase DUET's effectiveness in optimizing data mixtures for unseen evaluation tasks as compared to other data mixing or selection baselines.
___

We would like to summarize our responses to the reviewers during the rebuttal period. Specifically,

1. **Reviewer `6k1g`:** we re-emphasized the key novelty of DUET that extends beyond simply applying BO. Specifically, DUET is the first algorithm to interleave different data selection methods into Bayesian optimization (BO) to optimize data mixtures with strong theoretical and empirical justifications.
2. **Reviewer `6k1g`:** we provided qualitative analysis of data mixtures found at every iteration in DUET, clarifying their questions and technical misconceptions of the effectiveness of DUET and BO in general. **We added these analysis to App. D.3. of our revised manuscript.** We also pointed the reviewer to the real performance landscape we found in our paper (Fig. 2) and used it to clarify why DUET seems to perform well in just a few iterations.
3. **Reviewer `9jKt`:** we additionally ran experiments on **full-parameter fine-tuning of 8B and 14B LLM models** and showed that DUET outperforms plain data selection in optimizing data mixtures for unseen tasks. **We have included them into App. D.1 of our revised manuscript.** We also noted how IF-score computation can be scaled to larger models via the use of smaller surrogate models.
4. **Reviewer `9jKt`:** we clarified the efficiency of DUET by explaining how interleaving data selection with Bayesian optimization _reduces_ the need to brute-force iterate through every possible training data mixtures. DUET's ability to exploit feedback from a target task also aligns with how **real-world LLMs can afford to be improved through multiple rounds of feedback over sufficiently long periods of time in practice** instead of needing to obtain optimal data mixtures near-instantly.

5. **Reviewer `9jKt`, `QVFn`:** we presented some interesting insights into the use of Influence Function (IF) as a data selection choice in DUET. Namely, how IF scores are computed efficiently and robustly in practice, why they work effectively in DUET, and how DUET's regret bounds change with and without the IF-driven estimator.

6. **Reviewer `tBBS`:** we discussed DUET's computation and memory overhead, and how it scales with the number of data domains present in the training data. **We have included these details in App. D.2 of our revised manuscript.**

New additions can be found in App. D.1, D.2, D.3 (in _blue_) of our revised manuscript. We hope this summary can aid the AC's decision-making process, thank you.

Best regards,

Authors

---

### Meta-Review · Area_Chair_88sX · 2026-01-12

**Summary:**

1. The computational cost and overhead have not been clearly discussed;
2. The reliance on IF is questionable, as some of works point out IF for LLM is not very reliable;
3. The experiments are confined to fine-tuning. The approach's feasibility for large-scale pre-training, especially for computationally heavy estimators like IF, is not demonstrated;
4. The technical novelty is limited, as the proposed method is a simple combination of data selection and Bayesian optimization;

**Reviewer Concerns:**

Addressed concerns:

2. IF is reliable for identifying data points which are noisy and not relevant to the evaluation set at all, which useful for this setting.
3. The authors add experiments on full-parameter fine-tuning of 8B and 14B LLM models and showed that DUET outperforms plain data selection in optimizing data mixtures for unseen tasks.
4. The technical contributions lies in drawing the direct connection between how different data selection methods in DUET's inner loop influence Bayesian optimization's observation noise, and consequentially its regret. This concern is addmitted by reviewer QVFn.

Outstanding concerns:
1. The influence scores directly requires $O(Np^2 + p^3)$  operations ($N$ data points and $p$ model parameters), and the computation overhead of DUET-IF can be sped up $O(T^3 + Np)$. The memory overhead of DUET-IF is $O(T^2 + p)$. Although the authors provide detailed discussion of the method, the expensivie cost concern still remains.

**Reviewer Scores:**

No

---

### Decision · Program_Chairs · 2026-01-26

Accept (Poster)